# Structural insights into the agonists binding and receptor selectivity of human histamine H$_4$ receptor

Dohyun Im [1,9], Jun-ichi Kishikawa [2,9], Yuki Shiimura[1,3], Hiromi Hisano[1], Akane Ito[1], Yoko Fujita-Fujiharu [4,5,6], Yukihiko Sugita [4,5,7], Takeshi Noda[4,5,6], Takayuki Kato [2] ✉, Hidetsugu Asada [1] ✉ & So Iwata [1,8] ✉

Histamine is a biogenic amine that participates in allergic and inflammatory processes by stimulating histamine receptors. The histamine H$_4$ receptor (H$_4$R) is a potential therapeutic target for chronic inflammatory diseases such as asthma and atopic dermatitis. Here, we show the cryo-electron microscopy structures of the H$_4$R-G$_q$ complex bound with an endogenous agonist histamine or the selective agonist imetit bound in the orthosteric binding pocket. The structures demonstrate binding mode of histamine agonists and that the subtype-selective agonist binding causes conformational changes in Phe344[7.39], which, in turn, form the "aromatic slot". The results provide insights into the molecular underpinnings of the agonism of H$_4$R and subtype selectivity of histamine receptors, and show that the H$_4$R structures may be valuable in rational drug design of drugs targeting the H$_4$R.

Histamine, a biogenic neurotransmitter, exerts its pathophysiological functions in the central nervous system and peripheral tissues by stimulating histamine receptors belonging to the G protein-coupled receptor (GPCR) superfamily. Four histamine receptors (H$_1$R–H$_4$R) have been identified in humans, each with a different intracellular signaling mechanism[1]. In 1988 a Nobel prize was awarded for the discovery of H$_2$R antagonists, following the pharmacological discovery of H$_1$R and H$_2$R[2,3]. The antagonists of these receptors have been developed for the treatment of allergic diseases and gastric acid secretion, respectively, and are still in use[1,4]. H$_3$R was identified in the 80 s and has very low homology (<20%) with H$_1$R and H$_2$R[5,6]. It is a clinical target for central nervous system diseases, as it mainly regulates histamine function in the brain, and the H$_3$R antagonist pitolisant has been used for the treatment of narcolepsy (Wakix®) and obstructive sleep apnea (Ozawade®)[7].

H$_4$R is the last histamine receptor to be identified[8–12]. Since H$_4$R is mainly expressed in hematopoietic cells, such as monocytes, basophils, dendritic cells, and T-lymphocytes, it has been reported to participate in the release of cytokines, and consequently plays an important role in anti-inflammatory and immunomodulatory activities[13,14]. Therefore, H$_4$R may be a target for chronic inflammatory diseases (CIDs), such as asthma, arthritis, and atopic dermatitis[15,16]. Preclinical and clinical studies involving H$_4$R antagonists have demonstrated its therapeutic efficacy against CIDs, making H$_4$R a promising therapeutic target[4]. H$_4$R shares 34% of its amino acid sequence identity with H$_3$R, and 50% in the transmembrane (TM) region. Because of their structural similarities, there are many ligands that have a high affinity for both receptors[17,18]. However, it is important to further explore selective H$_4$R ligands to facilitate the development of more effective CID therapies.

[1]Department of Cell Biology, Graduate School of Medicine, Kyoto University, Konoe-cho, Yoshida, Sakyo-ku, Kyoto 606-8501, Japan. [2]Institute for Protein Research, Osaka University, 3-2 Yamadaoka, Suita, Osaka 565-0871, Japan. [3]Institute of Life Science, Kurume University, Kurume, Fukuoka 830-0011, Japan. [4]Laboratory of Ultrastructural Virology, Institute for Life and Medical Sciences, Kyoto University, 53 Shogoin Kawahara-cho, Sakyo-ku, Kyoto 606-8507, Japan. [5]Laboratory of Ultrastructural Virology, Graduate School of Biostudies, Kyoto University, 53 Shogoin Kawahara-cho, Sakyo-ku, Kyoto 606-8507, Japan. [6]CREST, Japan Science and Technology Agency, 4-1-8 Honcho, Kawaguchi, Saitama 332-0012, Japan. [7]Hakubi Center for Advanced Research, Kyoto University, Kyoto 606-8501, Japan. [8]RIKEN SPring-8 Center, 1-1-1 Kouto, Sayo-cho, Sayo-gun, Hyogo 679-5148, Japan. [9]These authors contributed equally: Dohyun Im, Jun-ichi Kishikawa. ✉e-mail: tkato@protein.osaka-u.ac.jp; asada.hidetsugu.4s@kyoto-u.ac.jp; s.iwata@mfour.med.kyoto-u.ac.jp

In a previous study, we determined the crystal structure of the inactive form of $H_1R$ ($H_1R_{dox}$), which is bound to doxepin, a first-generation antihistamine[19]. In the earlier study, the binding mode of doxepin and the reason for the low selectivity of first-generation antihistamines were investigated. Furthermore, the binding mode and specificity of second-generation antihistamines were determined using a computational approach. Recently, the cryo-electron microscopy (cryo-EM) structure of active $H_1R$ ($H_1R_{his}$) bound to an endogenous agonist histamine has been reported[20]. The structure revealed details of histamine binding in $H_1R$. The activation mechanism of $H_1R$ was assessed further by examining the details of $G_q$ protein engagement.

Here, we report the cryo-EM structures of $H_4R$, which include $H_4R$ bound to an endogenous histamine ligand ($H_4R_{his}$) and $H_4R$ bound to the $H_3R/H_4R$ selective agonist imetit ($H_4R_{ime}$). We elucidate the specific ligand recognition mechanism of $H_4R$ by comparing the binding mode of two agonists with different pharmacological characteristics. Furthermore, these results may provide insights that facilitate rational drug design targeting the $H_4R$.

## Results

### Overall structures of the agonist-bound $H_4R$-$G_q$ complex

For the cryo-EM experiment, $H_4R$ and trimeric G proteins ($G\alpha_q$, rat $G\beta_1$, and bovine $G\gamma_2$) were co-expressed using *Spodoptera frugiperda* (Sf9) insect cells. $H_4R$ is known to primarily engage with $G\alpha_i$ and transduce $G_i$ signals. We co-expressed and purified $G_i$-coupled $H_4R$ but failed to generate the complex (Supplementary Fig. 1b). However, the complex was successfully obtained by using the non-canonical miniGqiN as an accessory protein for structural determination. Furthermore, $H_4R$ has been reported to have a weak $G_q$ signal ($pEC_{50} = 6.8$, Emax = 9% for $G_q$/$pEC_{50} = 7.1$, Emax = 26% for $G_i$)[21]; therefore, the $H_4R$-$G_q$ complex was used to determine the $H_4R$ structure.

The $H_4R$-$G_q$ complex structures, $H_4R_{his}$ and $H_4R_{ime}$, which were bound to histamine and imetit, were determined by a cryo-EM single-particle analysis at 3.0 Å and 3.1 Å resolutions, respectively (Fig. 1, Supplementary Fig. 2, and Table 1). In the $H_4R_{his}$ and $H_4R_{ime}$ structures, the side chains of most of the amino acids in the receptor and G protein regions could be assigned to the final EM map and refined with excellent geometry. Both the agonists used in the present study, histamine and imetit, were clearly identified, and a model was obtained to explain the recognition mechanisms of the agonists in $H_4R$ (Supplementary Fig. 3). In the overall structure of the $H_4R$-$G_q$ complex, similar to other GPCR-G protein complex structures, the $G_q$ trimer bound to the intracellular region of $H_4R$, and Nb35[22] and scFv16[23] bound to $G_q$ and the $G_q$-$G\beta$ interface, respectively, thereby stabilizing the complex (Fig. 1a, b, d, e). A local resolution analysis showed that the WD40 repeat core of the $G\beta$ subunit, N-terminal region of the $G\alpha$ subunit ($G_\alpha HN$), and scFv16 had the highest resolution, whereas the extracellular region of the receptor and a portion of the N-terminal region of $G\beta\gamma$ had a lower resolution (Supplementary Figs. 2e, f). In addition, the N/C-terminus, the intracellular loop 3 (ICL3), and some parts of the extracellular loops (ECL2, ECL3) of $H_4R$ were difficult to observe due to their high flexibility, and therefore, could not be modeled.

The $H_4R_{his}$ and $H_4R_{ime}$ structures exhibited a canonical GPCR scaffold with seven transmembrane helices (TM1–7) and an intracellular amphipathic helix 8 (H8) (Fig. 1c, f, Supplementary Fig. 3c). The root mean square deviation of both structures was 0.32 Å for the overall structure and 0.60 Å for the receptor region, indicating similar conformations. Histamine and imetit occupied an orthosteric binding pocket in the middle of the TMs, which was covered by a C-terminal segment of ECL2 stabilized by the disulfide bonds of $C87^{3.25}$ and $C164^{ECL2}$, which are common in other class A GPCRs (Fig. 1c, f).

**Table 1 | Cryo-EM data collection, refinement, and validation statistics**

| | Histamine-H4R-Gq complex (EMDB-33785) (PDB 7YFC) | Imetit-H4R-Gq complex (EMDB-33786) (PDB 7YFD) |
|---|---|---|
| **Data collection and processing** | | |
| Magnification | 81,000 | 81,000 |
| Voltage (kV) | 300 | 300 |
| Electron exposure (e–/Å²) | 50 | 50 |
| Defocus range (μm) | −0.8 to −2.0 | −0.8 to −2.0 |
| Pixel size (Å) | 0.88 | 0.88 |
| Symmetry imposed | C1 | C1 |
| Initial particle images (no.) | 1,823,691 | 2,410,519 |
| Final particle images (no.) | 180,728 | 628,467 |
| Map resolution (Å) | 3.0 | 3.1 |
| FSC threshold | 0.143 | 0.143 |
| **Refinement** | | |
| Initial model used (PDB code) | 7DFL, 7F9Z | 7DFL, 7F9Z |
| Model resolution (Å) | 3.3 | 3.2 |
| FSC threshold | 0.5 | 0.5 |
| Map sharpening B factor (Å²) | −108.22 | −134.07 |
| Model composition | | |
| Non-hydrogen atoms | 9882 | 9934 |
| Protein residues | 1262 | 1260 |
| Ligands | HSM; 1, CLR: 1 | IME: 1, CLR: 2 |
| B factors (Å²) | | |
| Protein | 58.9 | 37.7 |
| Ligand | 64.3 | 64.9 |
| R.m.s. deviations | | |
| Bond lengths (Å) | 0.005 | 0.003 |
| Bond angles (°) | 0.656 | 0.534 |
| Validation | | |
| MolProbity score | 1.44 | 1.57 |
| Clashscore | 5.92 | 7.16 |
| Poor rotamers (%) | 0.09 | 0.19 |
| Ramachandran plot | | |
| Favored (%) | 97.42 | 97.01 |
| Allowed (%) | 2.58 | 2.99 |
| Disallowed (%) | 0 | 0 |

### Molecular basis for the recognition of histamine and imetit by $H_4R$

Histamine and imetit share a common imidazole backbone, and imetit is characterized by a more extended structure with an additional isothiourea group, when compared with histamine (Fig. 1c, f). In the cryo-EM structure, histamine and imetit shared a common ligand-binding pocket surrounded by residues consisting of TM3, 4, 5, 6, and 7 (Fig. 2a, b, d, e). The binding mode of histamine and imetit was verified by a transforming growth factor (TGF)-α shedding assay using mutants of $H_4R$, and was observed to be consistent with that in the model structure (Fig. 2c, f, Supplementary Fig. 4 and Supplementary Table 1)[24].

In the structure of $H_4R_{his}$, the primary amino group of the ethylamine portion of histamine formed a salt bridge with $Asp94^{3.32}$ (Fig. 2a). This interaction is widely conserved in aminergic receptor structures, including the $H_1R$ structure. A loss of activity was observed in the $D94^{3.32}A$ mutant of $H_4R_{his}$ (Fig. 2c). In many aminergic receptors,

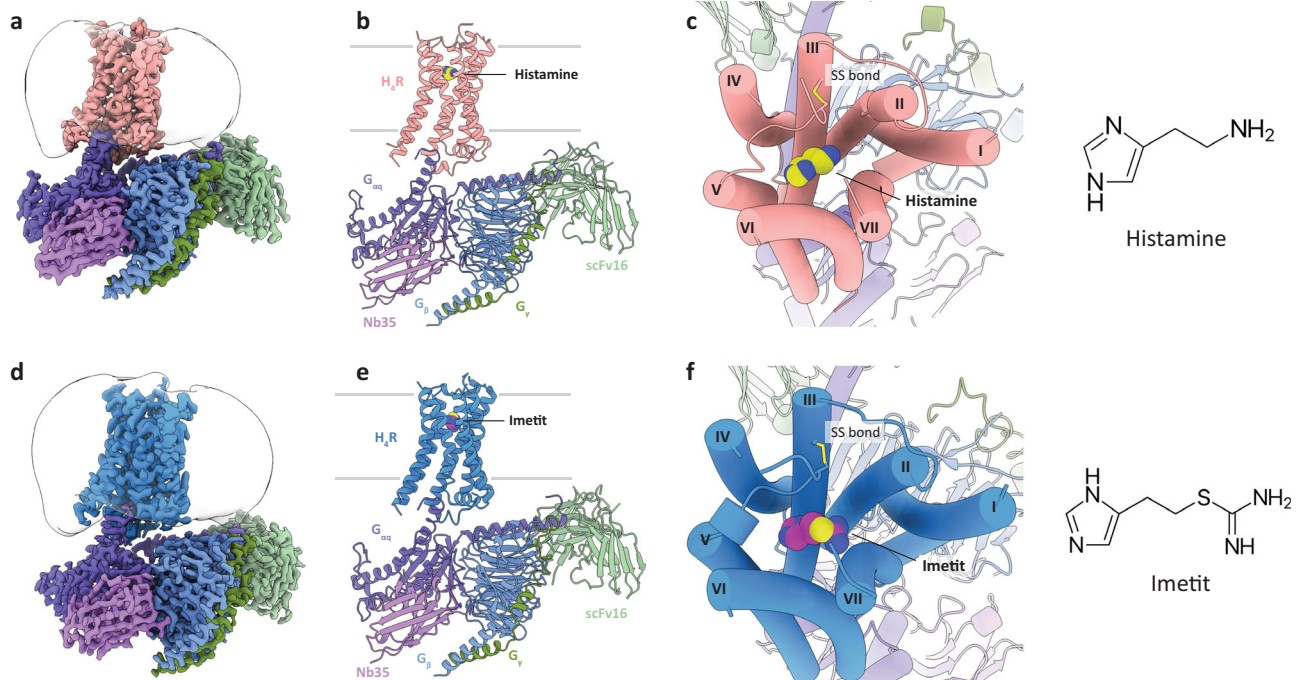

**Fig. 1 | Overall structures of the H4R signaling complexes. a, b** Orthogonal views of the cryo-EM density map (**a**) and model (**b**) of the histamine-bound H4R-Gq complex. **c** Extracellular view of the histamine-bound H4R-Gq complex and chemical structure of histamine. **d, e** Orthogonal view of the cryo-EM density map (**d**) and model (**e**) of the imetit-bound H4R-Gq complex. **f** Extracellular view of the imetit-bound H4R-Gq complex and chemical structure of imetit. Cys87[3.25] and Cys164[ECL2] form a disulfide bond and are shown as stick models (**c, f**). In (**a**)−(**f**), the complexes are colored by subunits. Histamine-bound H4R: pink, imetit-bound H4R: blue, Gq: purple, Gβ: navy, Gγ: green, Nb35: violet, scFv16: lime, histamine: yellow, and imetit: magenta.

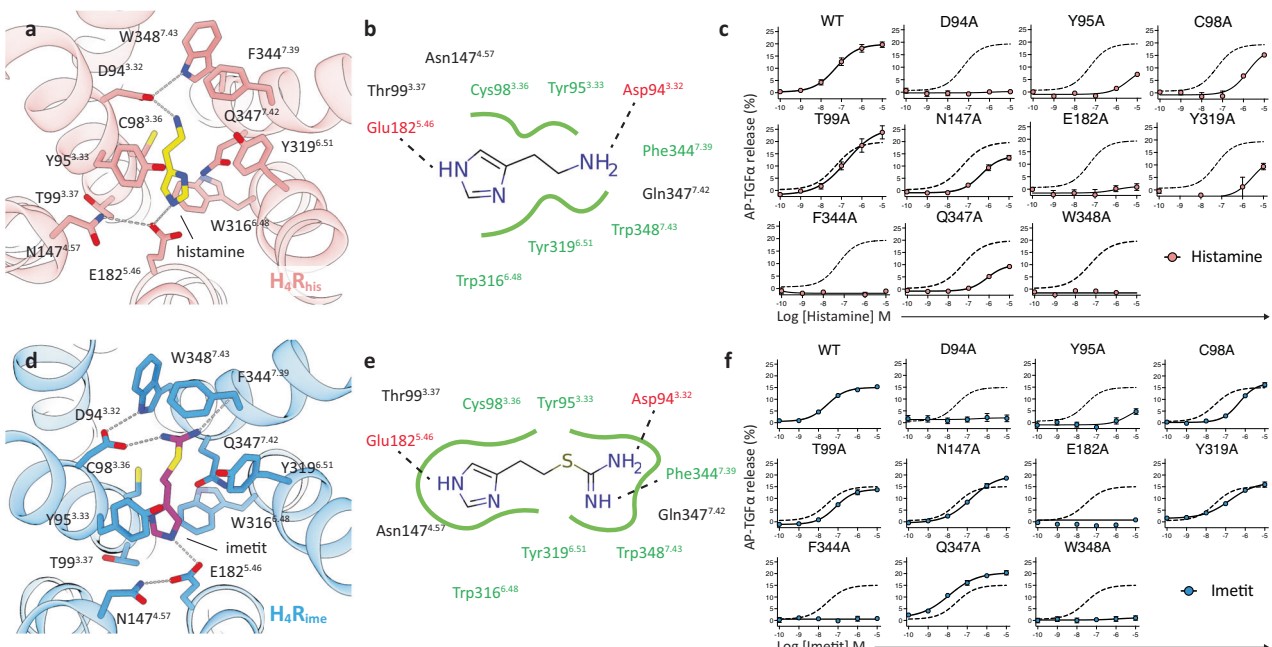

**Fig. 2 | Agonist recognition of H4R. a** Detailed interactions between histamine and H4R. Histamine and contact residues are shown as yellow and pink sticks. **b** Schematic representation of histamine-binding interactions. Hydrogen bonds are shown as dashed lines, and hydrophobic interactions and amino acids are shown in green. **c** Mutant study to assess the histamine response by the TGFα shedding assay. Dashed lines in the mutant graphs represent the wild-type (WT) H4R response. Concentration-response curves are shown as the mean ± standard error of the mean (SEM) from three independent experiments performed in triplicates. In many data points, the error bars are smaller than the symbols and are therefore not visible. **d** Detailed interactions between imetit and H4R. Imetit and contact residues are shown as magenta and blue sticks. **e** Schematic representation of imetit-binding interactions. **f** Mutant study to assess the imetit response by the TGFα shedding assay. Details are the same as in (**c**).

$Asp^{3.32}$ forms a hydrogen bond with the neighboring $Trp/Tyr^{7.43}$, thereby stabilizing its side chain and facilitating interactions with the amine group of the ligand[25]. Hydrogen bonding between $Asp94^{3.32}$ and $Trp348^{7.43}$ was also observed in $H_4R_{his}$, and the loss of signaling activity of the mutant $W348^{7.43}A$ indicated that the stability of $Asp94^{3.32}$ is important for ligand binding (Fig. 2a, c). The TM7 residues of $Phe344^{7.39}$, $Gln347^{7.42}$, and $Trp348^{7.43}$ are located within the van der Waals radius of the primary amino group of histamine and form hydrophobic interactions with it. Furthermore, $Gln347^{7.42}$ forms weak polar interactions with the imidazole group of histamine. These interactions are essential for histamine binding, and their importance was confirmed by the reduced activity of $Phe344^{7.39}$ and $Gln347^{7.42}$ in the Ala mutant (Fig. 2c).

On the opposite side of histamine from $Asp94^{3.32}$, $Tyr319^{6.51}$ is conserved in aminergic receptors in a similar manner as $Tyr/Phe^{6.51}$, suggesting that it is an essential residue for agonist and antagonist binding[19,20,26,27]. In $H_4R_{his}$, histamine forms hydrophobic contacts with this residue, and it was verified that the introduction of mutations to this residue is directly related to the decrease in receptor activity (Fig. 2a–c). $Tyr/Phe^{6.51}$ of TM6 plays an important role in ligand binding by forming aromatic clusters with $Trp^{6.48}$ and $Phe^{6.52}$ in the aminergic receptor[28]. In addition to ionic binding to $Asp^{3.32}$, many aminergic ligands form an aromatic stacking or hydrophobic interaction with this aromatic cluster. In $H_4R_{his}$, $Trp316^{6.48}$ was further coordinated to the bottom of the ligand binding pocket and exhibited a stacking interaction with the imidazole ring of histamine. So far, $H_4R$ mutant experiments have already been conducted on some of the above-mentioned amino acid residues in previous studies[29,30], and it was confirmed that there was no significant deviation from the experiments in the present study.

The imidazole ring of histamine was bound to a pocket consisting of $Tyr95^{3.33}$, $Thr99^{3.37}$, $Asn147^{4.57}$, $Glu182^{5.46}$, $Trp316^{6.48}$, and $Tyr319^{6.51}$. As mentioned above, the imidazole ring was stabilized by pi–pi stacking interactions from both the upper and lower sides, with $Tyr95^{3.33}$ as a lid, in addition to $Trp316^{6.48}$ at the bottom of the pocket (Fig. 2a, b). In particular, the $Y95^{3.33}A$ mutant dramatically reduced histamine activity, suggesting that it plays an essential role in binding (Fig. 2c). In addition, $Glu182^{5.46}$ is an important residue in the binding of the imidazole ring. The nitrogen atom ($N^{\tau}$) on the third position of the imidazole ring formed an ionic pair with $Glu182^{5.46}$, suggesting the importance of $Glu182^{5.46}$ along with that of $Asp94^{3.32}$ in the binding process. Previous studies have reported that mutants of this residue prevent histamine binding. This is in line with our signaling activity data (Fig. 2c)[29,30]. Although $Thr99^{3.37}$ and $Asn147^{4.57}$, which form a pocket, do not interact directly with the ligand, the signaling activity of the $T99^{3.37}A$ and $N147^{4.57}A$ mutants was found to be 2.4- and 9.2-fold lower, respectively, than that of wild-type $H_4R$. $Thr99^{3.37}$ is involved in pocket formation, and $Asn147^{4.57}$ contributes to the stabilization of the interaction with histamine by forming a hydrogen bond with the neighboring $Glu182^{5.46}$ and regulating the coordination of its side chain, as observed between $Trp348^{7.43}$ and $Asp94^{3.32}$ (Fig. 2a–c, Supplementary Table 1). Position 4.57 is $Asn^{4.57}$ in human $H_4R$ but $His^{4.57}$ in pigs and dogs; therefore, altering the side chain conformation of $Glu^{5.46}$ has been reported to be a possible reason for the species-specific differences in histamine affinity[31].

In $H_4R_{ime}$, imitit was found to bind to a pocket in a manner similar to that of histamine (Fig. 2a, b, d, e). Imitit had two hydrogen bond donors, which were complementary to the two negative residues of the binding pocket, $Asp94^{3.32}$ and $Glu182^{5.46}$, respectively. Mutagenesis experiments confirmed that these interactions are essential for imitit binding (Fig. 2f). Previous studies have suggested that the isothiourea group is a key determinant of the $H_4R$ activity enhancement in histaminergic compounds[18]. The cryo-EM structure of $H_4R_{ime}$ revealed that the isothiourea group of imitit plays a significant role in ligand binding. This substructure of the agonist penetrated the subpocket that is

formed by mostly aromatic and hydrophilic residues, such as $Tyr319^{6.51}$, $Phe344^{7.39}$, $Gln347^{7.42}$, and $Trp348^{7.43}$ (Fig. 2d, e). The two nitrogen atoms existing in the region formed a salt bridge with $Asp94^{3.32}$ for the primary amino group and a hydrogen bond with the oxygen atom of the main chain of $Phe344^{7.39}$ for the methanimine group. The primary amino group not only interacted with $Asp94^{3.32}$, but also formed pi–cation interactions with $Tyr319^{6.51}$, $Phe344^{7.39}$, and $Trp348^{7.43}$, which constitute the hydrophobic pocket, thereby increasing ligand binding stability. Mutant $Y319^{6.51}A$ exhibited a 5.3-fold decrease in signaling, while mutants $F344^{7.39}A$ and $W348^{7.43}A$ had even greater effects when compared with wild-type $H_4R$, indicating that the residues are crucial for imitit binding (Fig. 2f, Supplementary Table 1). The imidazole ring of imitit was tightly accommodated in a pocket consisting of TM3, 4, 5, and 6 (Fig. 2d, e). Furthermore, the hydrophobic interactions formed between imitit and $Tyr95^{3.33}$, $Cys98^{3.36}$, $Thr99^{3.37}$, and $Asn147^{4.57}$ were shown to be crucial for its binding (Fig. 2f, Supplementary Table 1). Similar to histamine, the imidazole ring of imitit was stably coordinated in the pocket by forming stacking hydrophobic interactions with $Tyr95^{3.33}$ and $Trp316^{6.48}$. However, unlike histamine, the imidazole group of imitit does not interact with $Gln347^{7.42}$. This may be the difference between the two agonists in the $Q347^{7.42}A$ mutant (Fig. 2c, f, Supplementary Fig. 4b).

## Comparison of agonist-binding characteristics between $H_4R_{his}$ and $H_4R_{ime}$

Imitit was initially developed as an agonist for $H_3R$[32,33] but has also been reported to have a high binding affinity ($K_i$) for $H_4R$ (imitit $K_i = 2.7$ nM, histamine $K_i = 8.1$ nM)[10]. As described in the previous section, the residues constituting the orthosteric binding site in $H_4R_{his}$ and $H_4R_{ime}$ were almost identical. However, the binding mode of the two agonists was clearly different (Fig. 3, Supplementary Figs. 5a, b). Imitit has a larger surface area than histamine due to the substituted isothiourea moiety (histamine: 267.8 Å² vs. imitit: 372.3 Å²) (Supplementary Fig. 5c). Although $H_4R_{his}$ and $H_4R_{ime}$ possess ligand-binding pockets of approximately the same size, several unique recognition mechanisms were established to make each agonist acceptable. First, remarkable differences in the binding features of both agonists were observed in the pockets around TM6 and 7. The $H_4R_{ime}$ structure formed a distinct pocket (hereafter referred to as the "aromatic slot") consisting of $Tyr319^{6.51}$, $Phe344^{7.39}$, $Trp348^{7.43}$, and $Gln347^{7.42}$, which was not observed in the $H_4R_{his}$ structure. Compared to the histamine-bound structure, this pocket was generated by the Cβ–Cγ bond of $Phe344^{7.39}$ that rotated approximately 90 degrees, like a turning doorknob, allowing it to accept a more extended imitit (Fig. 3b, c). The isothiourea moiety of imitit is likely protonated and charged. Both nitrogen atoms are equivalent and actively utilize this specific pocket to form multiple interactions with residues in the pocket. This may contribute to the increased binding strength of imitit. In contrast, the primary amine of histamine interacts relatively simply with $Asp94^{3.32}$ and $Phe344^{7.39}$, while the isothiourea portion of imitit utilizes the entire aromatic slot, which may be one of the reasons why the binding affinity of imitit is approximately three times higher than that of histamine[10].

The interaction between imitit and the aromatic slot and the overall molecular geometry of imitit caused its imidazole ring to coordinate 1.8 Å closer to TM3 than histamine. Imitit further exhibited a stronger interaction with peripheral residues such as $Cys98^{3.36}$ and $Thr99^{3.37}$, which may have also contributed to the binding affinity of imitit for $H_4R$ (Fig. 3a). The formation of a distinct ligand-specific binding pocket by the conformational change of the Phe side chain, as observed in the aromatic slot of $H_4R$, has also been reported for the extended binding pocket in the crystal structure of the dopamine $D_2$ receptor (in this case, $Phe^{3.28}$), which is a similar aminergic receptor[27]. Since it is difficult to consider the structural plasticity of ligand-binding pockets in drug design, identifying the flexible ligand-binding pocket

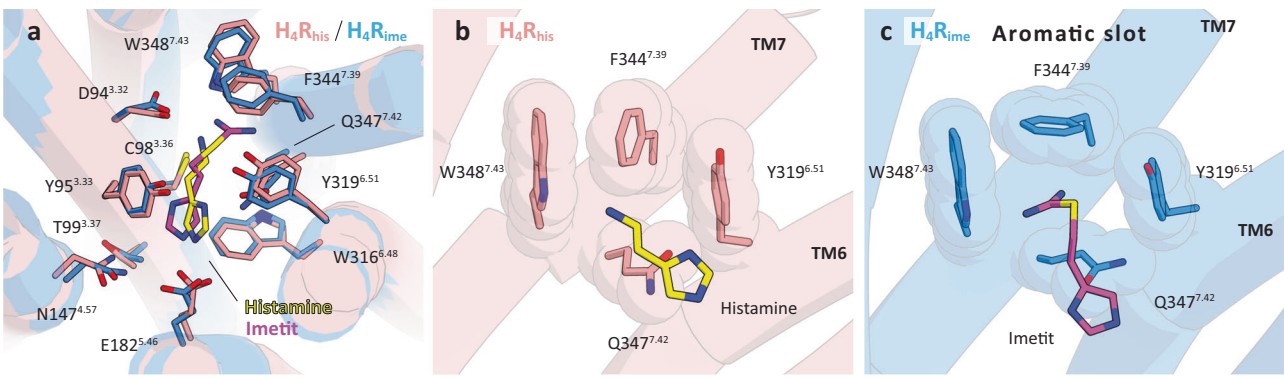

**Fig. 3 | Comparison of the binding poses between histamine and imetit for H4R.**
**a** Structural alignment of the agonist-binding pocket of H4R_his (pink) and H4R_ime (blue). Histamine (yellow), imetit (magenta), and the surrounding residues of each agonist are shown as sticks. **b, c** Hydrophobic residue cluster of TM7 and 8 in H4R_his (**b**) and H4R_ime (**c**). The side chain rearrangement of Phe344[7.39] forms the aromatic slot on H4R_ime. Residues comprising the pocket are indicated by sticks and transparent spheres and are pink (H4R_his) and blue (H4R_ime).

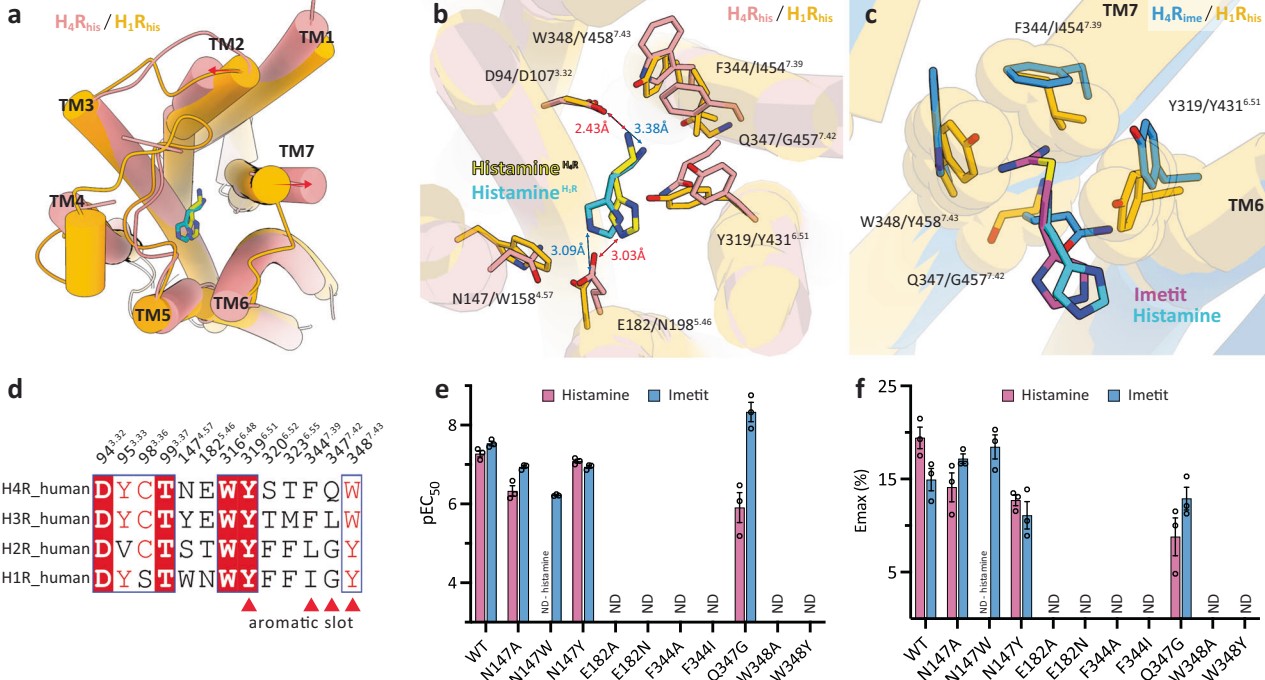

**Fig. 4 | Subtype selectivity in H4R and H1R. a, b** Structural superposition of H4R_his (pink) and H1R_his (orange, PDB 7DFL). (**a**) top view, (**b**) the agonist-binding pocket. The helices are shown as cylinders. The histamines of H4R and H1R are indicated by yellow and cyan sticks. **c** The aromatic slot on H4R_ime (blue) and the corresponding region for H1R_his (orange). Imetit and histamine are shown as magenta and cyan sticks. Residues comprising the pocket for H4R_ime are indicated by sticks and for H1R_his are shown as sticks and transparent spheres. **d** Sequence alignment of residues important for agonist binding in the histamine receptor family. Red triangles indicate residues corresponding to the aromatic slots in H4R. **e, f** TGFα shedding assay results of WT H4R and H4R mutants activated by histamine and imetit. The activities of the agonists are identified as pEC50 (**e**). Average Emax values were determined from the "log (agonist) vs. response -variable slope (three or four parameters)" function in GraphPad Prism 9 software (GraphPad Software Inc., San Diego, CA) (**f**). Data represent the mean ± SEM from $n = 3$ biologically independent experiments performed in triplicate. ND, not detected.

shown in our structure may make a significant contribution to rational drug design.

## Subtype-selectivity of H4R and H1R

While the affinity of H1R and H2R for histamine is in the μM range, H3R and H4R are known to have a high affinity in nM range[1,34]. Histamine receptor subtypes therefore require different concentrations of histamine for activation; however, the molecular basis for the selectivity between these subtypes remains unknown given the lack of structural information on histamine receptors. Here, we assessed the selectivity between subtypes in histamine receptors by comparing the cryo-EM

structure H1R_his of histamine-bound H1R[20] with the structures H4R_his and H4R_ime obtained in the present study.

The overall structure of the receptor region of H1R_his and H4R_his was relatively similar except for the extracellular ends of TM2 and 7 and the positions of the orthosteric binding pocket to which histamine bonds were almost identical (Fig. 4a). However, some of the residues that formed the ligand-binding pocket were different, which may have determined the affinity of histamine for both receptors. The most significant difference between the two receptors in histamine binding was the amino acid corresponding to position 5.46. Asn192[5.46] of H1R_his and Glu182[5.46] of H4R_his formed hydrogen bonds with Nτ atoms in the

imidazole ring of histamine; however, in $H_4R$, histamine exerted a stronger interaction with negatively charged Glu182[5.46] by establishing an ionic association (Fig. 4b). In contrast, the corresponding amino acid Asn192[5.46] in $H_1R$ is a neutral amino acid; therefore, the interaction was relatively weak, and the binding energy may have been lower than that in $H_4R$. As described in the previous section, Glu182[5.46] in $H_4R$ had its side chain coordination stabilized by hydrogen bonding with the neighboring Asn147[4.57], whereas position 4.57 in $H_1R$ was Trp and could not form a hydrogen bond network with position 5.46. The side chain stabilization effect was therefore not achieved. These structural differences were thought to affect the stability of the interaction with the $N^\tau$ atom of histamine. In fact, the $H_1R$ mimic mutants E182N[5.46] and N147W[4.57] showed a loss of signaling activity for histamine, while the $H_3R$ type mutant N147Y[4.57] at position 4.57 exhibited relatively similar $EC_{50}$ compared to wild type $H_4R$ (WT 55.4 nM vs. mutant 82.9 nM), as verified by the TGFα shedding assay. This indicated that the configuration and interaction of the residues in positions 4.57 and 5.46 in histamine binding are some of key factors that influence the difference in affinity between $H_1R$ and $H_4R$ (Fig. 4e, f, Supplementary Fig. 4 and Supplementary Table 1).

The primary amino group in histamine also exhibited different binding for both receptors. The ionic interaction at position 3.32, which is common in aminergic receptors, was conserved in both $H_1R_{his}$ and $H_4R_{his}$. In $H_1R_{his}$, however, the cationic amine moiety of histamine interacted relatively simply with Asp107[3.32] and Tyr431[6.51], whereas in $H_4R_{his}$, it interacted with the hydrophobic residue cluster on TM7 to gain greater binding affinity (Fig. 4b). Position 7.39 differed between $H_1R$ and $H_4R$ based on Ile and Phe, respectively, and comprised Phe344[7.39] in $H_4R$, which has a bulkier side chain that was present within a distance where it could interact with histamine. The $H_1R$ mimic mutant F344I[7.39] did not exhibit histamine activity, which indicated that this residue plays a critical role in histamine recognition (Fig. 4e, f). We posit that the interaction between histamine and Gln347[7.42] also affects histamine affinity. As previously mentioned, in the $H_4R_{his}$, Gln347[7.42] forms a polar interaction with the imidazole ring of histamine. However, such binding is not observed in $H_1R_{his}$ and the signaling activity of the $H_1R/H_2R$ type mutant Q347G[7.42] decreases about 46 times compared to that of the wild type (Fig. 4e, f, Supplementary Fig. 4 and Supplementary Table 1). Based on such observations, we consider the interaction of Gln347[7.42] and histamine in $H_4R$ to be one of the reasons for the high affinity.

Imetit is an $H_3R/H_4R$ selective agonist that does not bind to $H_1R/H_2R$ and has a high binding affinity for $H_3R/H_4R$ of 0.3 and 2.7 nM, respectively[10,33]. Since the publication of the structure of the first histamine receptor, doxepin-bound human $H_1R$[19], various computational chemistry approaches have been used to determine the molecular determinants of $H_3R/H_4R$ ligand binding, and several binding modes have been predicted. However, in most reports, no determinant other than Glu182[5.46] has been found to be important for subtype selectivity[35,36]. In the present study, we determined the structure of imetit-bound $H_4R$, which can be compared with $H_1R_{his}$, to assess the mechanisms underlying subtype selectivity in histamine receptors. As mentioned above, a detailed investigation of histamine binding to $H_1R/$ $H_4R$ revealed that Glu182[5.46] is undoubtedly one of the key residues involved in the selectivity of histamine receptors (Fig. 4b–f). Moreover, we identified the aromatic slot, a pocket that is crucial for the binding of the isothiourea region of imetit, as an important region for the selectivity of $H_1R$ and $H_4R$ (Fig. 4c). The aromatic slot consisting of Tyr319[6.51], Phe344[7.39], Gln347[7.42], and Trp348[7.43] was formed by the flexible orientation of the side chain of Phe in position 7.39, which is characterized by a mutually reinforced shape (Fig. 3b, c). The most important factor for the correct function of the aromatic slot is the flexibility of the side chain at position 7.39. In $H_3R/H_4R$, the residues corresponding to this position are both Phe, but in $H_1R/H_2R$, they are Ile and Leu, respectively. The formation and maintenance of the

aromatic slot by conformational changes in these side chains are therefore unlikely (Fig. 4c, d). Furthermore, the orientations of Ile454[7.39] and Tyr458[7.43] in $H_1R$ are likely to cause steric hindrance with the two nitrogen atoms in the isothiourea group. The acceptance of the isothiourea group of imetit in $H_1R$ is further likely to be challenging given the restricted coordination of the Tyr458[7.43] side chain in maintaining the Asp[3.32]–Trp/Tyr[7.43] interaction that is common in aminergic receptors.

The importance of the aromatic slot in the selectivity of histamine receptors was also verified in mutant experiments. $H_1R$ mimic mutants F344I[7.39] and W348Y[7.43] exhibited a loss of imetit activity, thereby supporting the above structural findings (Fig. 4e, f). However, Q347G[7.42] exhibited a slight increase in the imetit signal, which is consistent with the activity for Q347A[7.42] mutant mentioned earlier. The conformation of the extended side chain of Gln347[7.42] is thought to restrict the conformation of Trp316[6.48], and the removal of this restriction by mutation of Gln347[7.42] gives the Trp316[6.48] rotamer more flexibility, suggesting that it may interact more closely with imetit. Since the interaction between Gln347[7.42] and the imidazole ring of histamine was observed, we believe that its activity is reduced by the same mutations (Fig. 4e, f). We also found that this aromatic slot comprised a unique topology, not only for histamine receptors, but also for other aminergic receptors (Supplementary Fig. 6). The fact that residues in positions 7.39 and 7.42 of $H_4R$ were less conserved in other receptors supports that this sub-pocket in $H_4R$ is highly distinctive, and targeting this aromatic slot in $H_4R$ may be useful for the design of compounds specific to $H_4R$ (or $H_3R$). Although the isothiourea group of imetit did not appear to be a substituent developed for the aromatic slot, our structure indicated that it was a substitution that could effectively exploit the slot.

As mentioned in the previous section, aromatic clusters on TM6 (Trp[6.48], Phe/Tyr[6.51], and Phe[6.52]) play an important role in the binding of agonists and antagonists at aminergic receptors[25,28]. In $H_1R_{his}$, the imidazole ring of histamine also interacted with this aromatic cluster and formed a hydrophobic interaction with Phe435[6.55] within the Van Der Waals radius (Supplementary Fig. 7c). However, in $H_4R$, some of the residues constituting this aromatic cluster were compact polar residues (Ser320[6.52] and Thr323[6.55]) and created a wider polar sub-pocket between TM5 and 6 (Supplementary Fig. 7a, b). Since the residues in this position were distinguishable between $H_1R/H_2R$ and $H_3R/H_4R$, the effective utilization of this pocket with aromatic substituents may have been important for generating selectivity in histamine receptor subtypes (Fig. 4d).

## Activation mechanism of $H_4R$

Since the inactive structure of $H_4R$ has not yet been elucidated, we assessed the activation mechanism of $H_4R$ by comparing the inactive structure of antagonist-bound $H_1R$ ($H_1R_{dox}$)[19] with our agonist-bound $H_4R$ structure $H_4R_{ime}$. Four motif structures (CWxP, PI(V)F, DR(E)Y, and NPxxY) that are important for GPCR activation are conserved in $H_4R$; therefore, it is valid to compare $H_4R_{ime}$ and inactive $H_1R_{dox}$ to assess these motif structures and the activation mechanism of $H_4R$ (Fig. 5a). Different pharmacological features exist for each ligand that binds to the GPCR. This is thought to be the result of the orientation of the side chains of this conserved local microswitch that affects the global movement of the helical backbone[37,38]. In agonist-bound $H_4R_{ime}$, a hydrophobic interaction between the agonist and Trp316[6.48] (known as the "toggle switch") of TM6 was observed and caused a downward swing in the side chain (Fig. 5b). This conformational change in Trp316[6.48] promoted the rearrangement of the unique configuration of Pro186[5.50]-Val102[3.40]-Phe312[6.44] (PIF motif in other GPCRs) and Asp111[3.49]-Arg112[3.50]-Tyr113[3.51] (DR(E)Y motif), thus allowing movement outside the cytoplasmic end of TM6 (Fig. 5c, d, f). The NPxxY motif at the cytoplasmic end of TM7 also showed a conformational change in the side chain rotamer of Tyr358[7.53]. This conformational change

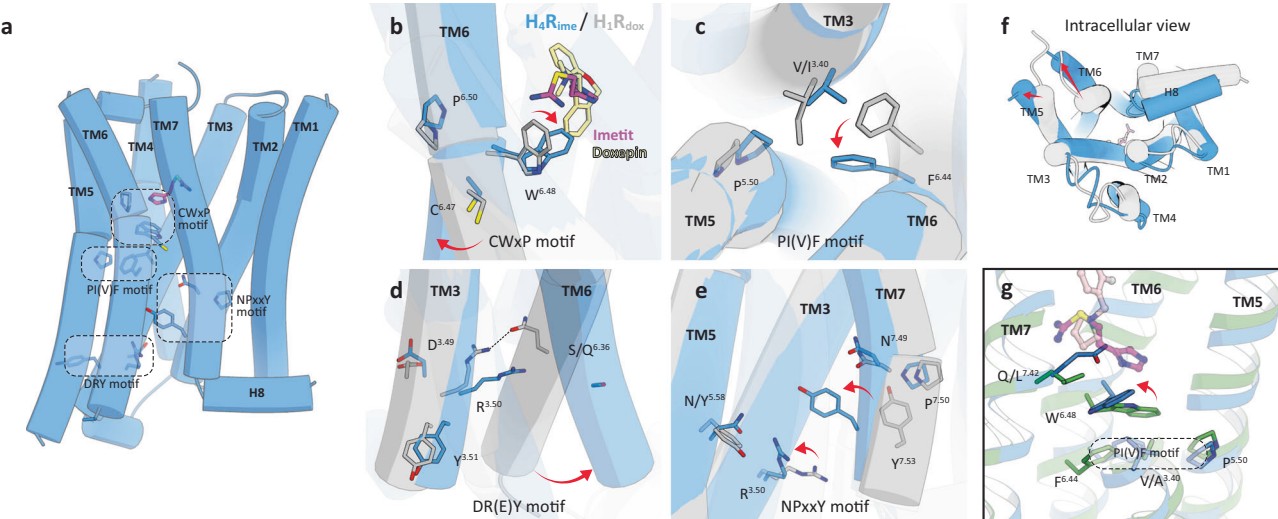

**Fig. 5 | Activation mechanism of H4R. a** Activation motifs in the H4R structure. Each motif residue is shown as a blue stick. **b–g** Structural comparison between the active H4Rime (blue) and antagonist-bound inactive H1Rhis (gray, PDB 3RZE). (**b**) CwxP motif, (**c**) PI(V)F motif, (**d**) DR(E)Y motif, (**e**) NPxxY motif. Imetit and doxepin are shown as magenta and khaki sticks. The residues involved in receptor activation are indicated by sticks, and conformational changes are shown as red arrows. Microswitch rearrangement allowed movement outside the cytoplasmic ends of TM5 and 6 (**f**). **g** Gln347[7.42] coordination and its effect on surrounding residues.

moved TM7 toward the TM3 and 5 sides, a distance that allowed an interaction with the DR(E)Y motif (Fig. 5e). Thus, the conformational changes of the microswitch and global movement of the TM bundle, which are commonly observed in GPCRs, were conserved in our structure. This confirmed that our H4Rhis and H4Rime structures are the active conformations that interact with trimeric G proteins.

Recently, the inactive structures of the antagonist-bound histamine H2 receptor (H2R) and the histamine H3 receptor (H3R) were published[39,40], and comparison of our agonist-bound H4R structure with the PF-03654746-bound inactive H3R structure suggests that the activation mechanism in H3R/H4R is different from that in H1R. As mentioned above, in the case of many aminergic receptors, the agonist interacts with the toggle switch, causing a downward swing of its side chain and rearrangement of the downstream structure to the active form. In the case of H3R/H4R, however, the present structure shows that the side chain of the toggle switch is not moved perpendicularly but horizontally (Fig. 5g). In H4R, the agonist formed the hydrophobic interaction with Trp316[6.48] and changes its side chain about 90° to the TM3 side, which may serve as a trigger in receptor activation. Here, the residue at position 7.42 may play an important role in controlling the conformational change of the toggle switch horizontally. The side chain of Gln347[7.42] in the cryo-EM structure of H4R is extended to the TM6 side, which is consistent with Leu401[7.42] conformation in the H3R structure. In most aminergic receptors, position 7.42 is conserved in Gly (only muscarinic acetylcholine receptors have Cys), but only H3R/H4R have Leu/Gln, which have larger side chains than Gly (Supplementary Fig. 8). In other words, the toggle switch within the H3R/H4R appears to have a unique movement, characterized by a lateral movement that maintains the "knock over" state to prevent steric hindrance with Leu/Gln[7.42].

## Discussion

In this study, we present the cryo-EM structures of the histamine H4 receptor bound by two agonists histamine and imetit, respectively, which exhibit distinct pharmacological properties. Although Gq signal activity could not be measured directly in the in-house measurement, weak Gq coupling was reported in the previous study[21]. The tethering system used in the present study effectively brings GPCRs and trimeric G proteins into close proximity, which is thought to make the binding with Gq stronger, enabling structural analysis.

In histamine receptor subtypes, H4R has low sequence similarities with H1R and H2R but has a more similar sequence identity with H3R. Given the overlap of the sequences, it would be difficult to develop a ligand that is H4R-selective and H3R-nonselective. Since the active structure of H3R is currently unknown, we generated a homology model of active state H3R based on the H4R structure revealed in the present study and compared their ligand recognition sites (Supplementary Fig. 9). Although the orthosteric binding sites in both receptors were expected to be similar, there were differences in some of the residues involved in ligand recognition (Fig. 4d, Supplementary Figs. 9a–c). Histamine had almost the same affinity for H3R/H4R, but OUP-16, which is an imidazole derivative agonist, exhibited 16-fold selectivity for H4R (H3R Ki = 2 μM / H4R Ki = 125 nM) (Supplementary Fig. 10)[15]. To investigate the difference in selectivity, we also performed a docking simulation of OUP-16 for the cryo-EM structure of H4R obtained in the present study (Supplementary Fig. 9d). Comparing the docking result of OUP-16 and homology model of H3R may explain the differences in their affinity. The docking simulation indicated that the guanidine moiety of OUP-16 engages in hydrogen bonding with Asp94. The imidazole ring on the opposite side of the agonist exhibits an alternative binding mode compared to the imidazole ring of histamine and is oriented toward the extracellular ends of TM5 and 6. Furthermore, the hydrogen bond with Glu182 is maintained and the interaction with Ser320 is established. The entire imidazole ring is occupied within the sub-pocket formed between TM4, 5, and 6. Within this pocket, however, there were different residues (Y/N[4.57], T/S[6.52], M/T[6.55]) for H3R/H4R, which may have resulted in the differences in pocket shape and properties in the two receptors (Supplementary Fig. 9a–c, d). The coordination of the hydrophobic and massive residues Tyr167[4.57] and Met378[6.55] next to Glu206[5.46], which is essential for hydrogen bonding with histamine in H3R, implies that the introduction of an additional sub-pocket is not tolerated. Therefore, the design of ligands that take into account the shape of the orthosteric binding pocket specific to H3R and H4R would allow for their respective selectivity to be generated. JNJ-7777120 was the first H4R-selective antagonist to be discovered and has contributed significantly to our understanding of the pathophysiological functions of H4R (Supplementary Fig. 10)[41]. In addition, the compound has interesting pharmacological characteristics, such as biased activity toward β-arrestin[42,43]. Several antagonists derived from this compound have

been demonstrated to be effective against pruritus, dermatitis, asthma, and arthritis in preclinical studies in animal models. Clinical trials have been conducted in some cases[44,45]. Since ligands that achieve such pharmacological effects require a high selectivity for $H_4R$, the structures of $H_4R$ revealed in the present study may provide information that could facilitate the development of $H_4R$-selective ligands. We also performed a docking study on JNJ-7777120 (Supplementary Fig. 9e). The results show that the $H_4R$-specific sub-pocket provides space to accept the chloroindole ring of JNJ-7777120, which is not present in other histamine receptors, including $H_3R$, and is expected to be an important region that allows subtype selectivity to be acquired. In addition, mutant experiments in previous reports have demonstrated that Asn147[4.57], Phe169[ECL2], Leu175[5.39], and Ser179[5.43], which correspond to this sub-pocket, decrease the activity of JNJ-777120, which could support our structural considerations[31,46].

Acquiring the inactive structure of $H_4R$ is a future challenge that may yield a better understanding of the antagonist binding and the activation mechanism of $H_4R$. The structural, biological, and pharmacological results of the $H_4R$ obtained in this study provide details on the molecular mechanisms of agonism in $H_4R$ and key insights into the mechanism of selectivity in histamine receptor subtypes. A rational structure-based design for effective drugs against CIDs may be developed further using the structural information on $H_4R$ presented herein.

## Methods
### Constructs
The full-length human $H_4R$ (residues 1–390) was subcloned into a pFastBac1 plasmid with an N-terminal hemagglutinin (HA) signal peptide, a FLAG tag, and 8× His tag, followed by a thermostabilized apocytochrome $b_{562}RIL$ (bRIL) epitope[47] to facilitate protein expression and stability (Supplementary Fig. 1a). An HRV 3 C protease cleavage site and linker residues were inserted between the $H_4R$ and bRIL. The artificial $G_q$ used in the present study was based on the mini-$G_s$ skeleton, which has two dominant-negative mutations and the alpha-helical domain truncation[48]. The 35 amino acids of the N-terminal were replaced with the corresponding sequence of $G\alpha_{i1}$ to facilitate scFv16 binding[49,50]. Such an approach was used to obtain the structures of $G_q$-bound 5-$HT_{2A}$ receptor/ghrelin receptor and $G_{11}$-bound $M_1$ receptors[51–53]. Furthermore, $G_i$ ($DNG_{i1}$) corresponding to Supplementary Fig. 1b was generated by introducing four dominant-negative mutations, S47N, G203A, A326S, and E245A[50]. To improve the stability and homogeneity of the receptor-G protein complex, we employed the NanoBiT tethering system with a large fragment of NanoLuc (LgBiT) and high affinity 11 amino acid peptide (HiBiT) tag attached to the C-terminus of the receptor and $G\beta_1$, respectively[54]. $G_q$, Nb35, scFv16, and Ric8A were cloned into the pFastBac1 plasmid. $His_{10}$-rat $G\beta_1$ with C-terminal HiBiT and bovine $G\gamma_2$ were subcloned into the pFastBac Dual vector.

### Protein expression and complex purification
Recombinant baculoviruses were prepared using the Bac-to-Bac baculovirus expression system (Invitrogen, Carlsbad, CA). *Spodoptera frugiperda* (Sf9) insect cells were grown to a density of 2–3 × $10^6$ cells/mL and co-infected with $H_4R$, $G_q$, $G\beta_1G\gamma_2$, Nb35, scFv16, and Ric8A viral stocks at a multiplicity of infection ratio of 8:2:2:1:1:1. They were harvested 48 h later. Cell pellets were resuspended in a buffer containing 20 mM of HEPES (pH 7.5), 100 mM of NaCl, 10% glycerol, 5 mM of $MgCl_2$, 5 mM of $CaCl_2$, 0.25 mM of TCEP, protease inhibitor cocktail (Nacalai Tesque), 25 mU/mL of apyrase, and 50 µM of $H_4R$ agonists (histamine or imetit) followed by incubation for 1 h at room temperature. Cell membranes were collected by centrifugation and solubilized in Solubilization Buffer containing 20 mM of HEPES (pH 7.5), 100 mM of NaCl, 10% glycerol, 5 mM of $MgCl_2$, 5 mM of $CaCl_2$, 0.25 mM of TCEP, protease inhibitor cocktail, 25 mU/mL of apyrase,

0.5% (w/v) lauryl maltose neopentyl glycol (LMNG, Anatrace) with 0.05% (w/v) cholesteryl hemisuccinate (CHS, Sigma-Aldrich), and 50 µM of agonists, for 2 h at 4 °C.

Insoluble materials were removed by centrifugation, and the supernatants were incubated with TALON metal affinity resin (Clontech) overnight at 4 °C. The resin was washed with 10 column volumes (CV) of Wash Buffer containing 20 mM of HEPES (pH 7.5), 100 mM of NaCl, 10% glycerol, 2 mM of $MgCl_2$, 2 mM of $CaCl_2$, 20 mM of imidazole, 0.25 mM of TCEP, 0.01% (w/v) LMNG, 0.01% (w/v) GDN (Anatrace), 0.001% (w/v) CHS, and 20 µM of agonists. The complex was then eluted with 3 CV of Elution Buffer containing 20 mM of HEPES (pH 7.5), 100 mM of NaCl, 10% glycerol, 2 mM of $MgCl_2$, 2 mM of $CaCl_2$, 250 mM of imidazole, 0.25 mM of TCEP, 0.01% (w/v) LMNG, 0.001% (w/v) GDN (Anatrace), 0.001% (w/v) CHS, and 20 µM of agonists. The N-terminal FLAG tag and bRIL epitope were cleaved by an HRV 3 C protease for 3 h at 4 °C. Finally, the complex was concentrated to 0.5 mL using the Amicon Ultra-15 concentrator (Millipore) and subjected to size-exclusion chromatography on a Superdex 200 Increases 10/300 column (GE Healthcare) pre-equilibrated with a buffer containing 20 mM of HEPES (pH 7.5), 100 mM of NaCl, 2 mM of $MgCl_2$, 2 mM of $CaCl_2$, 0.25 mM of TCEP, 0.0015% (w/v) LMNG, 0.0005 (w/v) GDN, 0.00015% (w/v) CHS, and 20 µM of agonists. The fractions containing the monomeric complex were collected and concentrated to approximately 10 mg/mL for electron microscopy experiments. Expression and purification of the $H_4R$-$G_i$ complex (corresponding to Supplementary Fig. 1b) was performed similar to that for $G_q$ above. However, since Nb35 does not bind to $G_i$, Nb35 co-expression was not performed.

### Cryo-EM grid preparation and data collection
The Quantifoil R1.2/1.3 holy carbon copper grid (Quantifoil) was glow-discharged at 7 Pa with 10 mA for 10 s using a JEC-3000FC sputter coater (JEOL) prior to use. A 3-µL aliquot was applied to the grid, blotted for 3.5 s with a blot force of 10 in 100% humidity at 8 °C, and plunged into liquid ethane using a Vitrobot Mark IV (Thermo Fisher Scientific). Cryo-EM data collection for screening sample quality and grid conditions was performed using a Glacios Cryo-transmission electron microscope operated at 200 kV accelerating voltage with a Falcon4 camera (Thermo Fisher Scientific) at the Institute for Life and Medical Sciences, Kyoto University. After several screening sessions, data were collected using a Titan Krios (Thermo Fisher Scientific) at 300 kV accelerating voltage equipped with a direct K3 electron detector, Gatan BioQuantum energy filter (slit width of 20 eV) (Gatan), and Cs corrector (CEOS, GmbH), which were installed at the Institute for Protein Research, Osaka University. Data collection was carried out using SerialEM software[55] at a norminal magnification of ×81,000 (calibrated pixel size of 0.88 Å pixel$^{-1}$) with a total exposure time of 5.0 s (50 frames) with a defocus range of −0.8 to −2.0 µm. The detailed imaging conditions are described in Table 1.

### Cryo-EM data processing
All image processing was performed using cryoSPARC[56]. We manually inspected and curated the micrographs after patch-based motion correction and contrast transfer function (CTF) estimation. The particles for the histamine-bound-$H_4R$-$G_q$ complex were selected using the Blob particle picker and initial two-dimensional (2D) classification yielded templates for subsequent template picking. A subset of the selected particles was used as a training set for Topaz, which was used to repick the particles from the micrographs[57]. The particles for the imetit-bound $H_4R$-$G_q$ complex were picked using the automated procedure in crYOLO[58]. The picked particles were subjected to a 2D classification to discard particles in poorly defined classes. Ab initio reconstruction was performed in cryoSPARC asking for four classes, which resulted in one good class and three trash classes. Multiple rounds of heterogeneous refinement were then performed against the

four ab initio models to remove bad particles. The selected particles were extracted at full pixel size and subjected to non-uniform (NU) refinement[59]. Further classification without alignment was performed using a soft protein mask to create a better-looking ligand binding region for the histamine-bound-H4R-Gq complex.

The densities were further Improved by per-particle defocus refinement and another round of non-uniform refinement to generate the final map. Resolutions were estimated using the 'gold standard' criterion (FSC = 0.143). The local resolution was calculated in cryoSPARC. Map sharpening was reevaluated with the Phenix autosharpen tool[60,61]. These maps were used for modeling. The validation on the coordinate refinement ($FSC_{work}/FSC_{test}$) was calculated using servalcat[62]. The processing strategy is described in Supplementary Fig. 2.

## Model building and refinement
Model building was facilitated by both the previous cryo-EM structures of the GHSR-Gq complex (PDB 7F9Z) and H1R bound to histamine (PDB 7DFL)[20,53]. The receptor, Gq trimer, Nb35, and scFv16 models were manually built in Coot[63], followed by several rounds of real-space refinement using Phenix[64]. All molecular graphics were prepared using CueMol (http://www.cuemol.org) and UCSF ChimeraX[65]. The 3D reconstruction and model refinement statistics are summarized in Table 1.

## Molecular docking
The docking study was performed using $H_4R_{ime}$ with the Glide program[66] in the Schrödinger Suite 2022-4 (Schrödinger LLC). The coordinates of OUP-16 and JNJ-7777120 were initially built using LigPrep in the Schrödinger Suite at pH 7.0 and the OPLS3 force field. Each ligand was docked into the binding pocket of H4R, resulting in ten conformations per ligand. The most reliable binding poses were selected based on the interaction energy and visual inspection.

## TGFα shedding assay
The mutants were prepared using the primers listed in Supplementary Table 2. The activity of the endogenous agonist (histamine) and its derivative (imetit) for H4R mutants on G protein signaling was determined using a TGFα shedding assay. In brief, a pCAGGS plasmid encoding the human wild-type or a mutant H4R (human, full-length, and untagged), together with pCAGGS plasmids that encoded the chimeric $G\alpha_{q/i1}$ subunit and alkaline phosphatase-tagged TGFα (AP-TGFα; human codon-optimized), were transfected into HEK293A cells by a Lipofectamine® 3000 (LFA) transfection reagent (Thermo Fisher Scientific) (125 ng of the H4R plasmid, 125 ng of the $G\alpha_{q/i1}$ plasmid, 625 ng of the AP-TGFα plasmid, and 18.2 μL of LFA per well in a six-well culture plate). The chimeric $G\alpha_{q/i1}$ subunit comprises the $G\alpha_q$ backbone and $G\alpha_{i1}$-derived six-amino acid C-terminus and couples with $G_i$-coupled H4R but induces a $G_q$-dependent TGFα shedding response[24]. After culturing for 1 day and incubating at 37 °C in a 5% $CO_2$ incubator, the transfected cells were harvested by trypsinization, neutralized with Dulbecco's modified Eagle's medium containing 10% fetal calf serum (FCS) and penicillin–streptomycin, washed once with Hank's balanced salt solution (HBSS) containing 5 mM of HEPES (pH 7.4), and resuspended in 6 mL of the HEPES-containing HBSS. The cell suspension was seeded into a 96-well plate at a volume of 90 mL per well (typically, 48 wells per transfected cells) and incubated for 30 min in the $CO_2$ incubator. Test compounds (10×, diluted in 0.01% BSA and 5 mM of HEPES-containing HBSS, 10 mL volume) were added to duplicate wells and incubated for 1 h. After centrifugation, the conditioned medium (80 mL) was transferred to an empty 96-well plate. AP reaction solution [10 mM of p-nitrophenylphosphate (p-NPP), 120 mM of Tris–HCl (pH 9.5), 40 mM of NaCl, and 10 mM of $MgCl_2$] was dispensed into the cell culture plates and plates containing conditioned media (80 mL). Absorbance was measured at 405 nm before and after a 1-h incubation period at 25 °C using a microplate reader (iMark™ Microplate Absorbance Reader; BioRad). Ligand-induced AP-TGFα release was calculated as previously described[24]. The vehicle-treated AP-TGFα release signal was set as a baseline unless otherwise specified. AP-TGFα release signals were fitted with a four-parameter sigmoidal concentration-response curve, from which $EC_{50}$ and $E_{max}$ values were obtained, using GraphPad Prism 9 software (GraphPad Prism). Negative values of logarithmically transformed $EC_{50}$ values ($pEC_{50}$) were used to calculate the mean and standard error of independent experiments.

To estimate the expression level of each H4R mutant, a FLAG tag was added to the N-terminus of all mutants. Transfection of the genes into the cells was performed as described above, and the FLAG-tagged mutant genes were transfected at the same time as all the genes required for the TGFα shedding assay. One day after gene transfer, transfected cells were harvested and stained with anti-DYKDDDDK tag monoclonal antibody (FUJIFILM Wako Pure Chemical Corporation) as primary antibody and Goat anti-mouse IgG antibody conjugated with Alexa Fluor™ 488 (Thermo Fisher Scientific) as a secondary antibody for 30 min on ice in the dark. Stained cells were replaced with FACS buffer (PBS(-) with 2% FCS and 0.1% $NaN_3$), and cells were assayed using a flow cytometer (Guava EasyCyte Plus, Millipore). The expression levels of the mutant were estimated in terms of the mean fluorescence intensity (MFI) ratio (MFI of the mutant compared to that of the WT (Supplementary Fig. 4a).

## Statistical analysis
All functional study data were analyzed using Prism 9 (GraphPad) and are presented as mean ± standard error of the mean (SEM) of three independent experiments. Statistical analyses were performed using Prism 9 (GraphPad) with one-way analysis of variance followed by Dunnett's post-hoc test. Values with $p < 0.05$ are considered statistically significant.

## Reporting summary
Further information on research design is available in the Nature Portfolio Reporting Summary linked to this article.

## Data availability
The cryo-EM density maps and atomic coordinates have been deposited in the Electron Microscopy Data Bank (EMDB) and wwPDB under accession numbers EMD-33785 and 7YFC [https://doi.org/10.2210/pdb7YFC/pdb] for the histamine-bound H4R-Gq complex, EMD-33786 and 7YFD [https://doi.org/10.2210/pdb7YFD/pdb] for the imetit-bound H4R-Gq complex. Raw movies were deposited to EMPIAR data base (https://www.ebi.ac.uk/empiar/) with accession numbers EMPIAR-11708. Source data are provided with this paper.

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

## Acknowledgements

This work was supported by a Grant-in-Aid from the Japanese Ministry of Education, Culture, Sports, Science and Technology (19H00923 (S.I.) and 23K06357 (D.I.)) and the Platform Project for Supporting Drug Discovery and Life Science Research (Basis for Supporting Innovative Drug Discovery and Life Science Research (BINDS)) from Japan Agency for Medical Research and Development (AMED) under the grant number JP21am0101079 and JP23ama121007 (S.I.). This study was also supported by the Takeda Science Foundation and the Mochida Memorial Foundation for Medical and Pharmaceutical Research (D.I.). This work was performed in part under the Collaborative Research Program as the Visiting Fellow of Institute for Protein Research, Osaka University, VFCR–23-02 (S.I.) and in part using the cryo-electron microscope under the Collaborative Research Program of Institute for Protein Research, Osaka University, CEMCR-23-02 (H.A.).

## Author contributions

D.I., H.A., and S.I. designed the experiments. D.I. prepared the cryo-EM samples, and Y. Shiimura helped with the construction of $G_q$-heterotrimer expression. D.I., J.K., Y.F.F., Y. Sugita, T.N., T.K., and H.A. performed the cryo-EM analysis. H.H., A.I., and H.A. performed the TGFα shedding assay. D.I., H.A., and S.I. wrote the paper with assistance from all of the authors. T.K., H.A., and S.I. supervised the project. All authors have read and approved the final version of the manuscript.

## Competing interests

The authors declare no competing interests.
