## [Peer Review File · Nature Communications]

Structural Insights into the Agonists Binding and Receptor Selectivity of Human Histamine H4 ReceptorREVIEWER COMMENTS

Reviewer #1 (Remarks to the Author):

This manuscript presents two cryo-EM structures of the human histamine H4 receptor bound to the endogenous agonist histamine and H3/H4 subtype selective agonist imetit in complex with engineered Gq protein, stabilized in detergent micelles by a NanoBiT tethering technology, a single chain antibody fragment scFv16, and a nanobody Nb35. The structures along with site-directed mutagenesis coupled to functional assays provided novel insights into the agonist binding mode, subtype selectivity and activation of histamine receptors. Previously, structures of the inactive and active states of the histamine H1 receptor were published. This work uncovers the first structure of the H3/H4 subtypes of the histamine receptor, which have low homology with H1/H2 subtypes, and therefore should provide considerable interest for researchers studying signaling mechanisms of the histamine receptors and for structure-based discovery of subtype selective tool compounds and drug candidates.

The structures appear to be of reasonably good quality and the study has been well executed, however, the manuscript suffers from multiple grammar issues making it difficult to read and understand. Below I provide few major and several minor critiques, addressing which should help with improving this manuscript.

Major comments

1. It is mentioned in the manuscript that H4 receptor couples primarily to Gi protein, however, the H4R-Gi complex was unstable, and a complex with an engineered Gq protein was used instead to determine the structure. Do you have any explanations why Gq worked better than Gi? Have you tried G16, which shows stronger signaling than Gq in Ref. 21. Please add units for EC50 and Emax values in Line 91 and also include corresponding data for Gi signaling for comparison.
2. Related to the first question, do you consider the obtained structure to be a genuine structure of a H4R-Gq complex or is it just an artificially constructed complex helping to obtain a structure of H4R bound to agonists? If it's a genuine complex, then the interactions between H4R and Gq should be described and compared to interactions between H1R and Gq. Do these structures provide any clues why H1R and H2R predominantly couple to Gq while H3R and H4R to Gi?
3. The toggle switch residue W316^{6.48} has different conformations in structures with imetit and histamine. Are these different conformations supported by different density? And if so, do they have any implications in the signaling mechanism?

4. Signaling data with histamine (Lines 332-335) do not really support the hypothesis about the basal activity. Additionally, the Q347A mutant increased signaling potency of imetit, which is not discussed in the manuscript. What should be compared here is the basal activity of these mutants with respect to the basal activity of the wild type receptor.

5. The Discussion section contains many speculations about H3R vs H4R selectivity that are not supported by any experimental or computer modeling data. This question is indeed interesting and important, and the author should design some specific mutations as well as try docking in H3R model vs H4R structure.

Minor comments

6. Line 89. Check grammar. "The complex was able to obtain..."

7. Line 95. Incorrectly stated that resolution reflects the model to map correlation. Here the model was not used; instead, stated resolution corresponds to FCS=0.14 between two half maps.

8. Line 100. "H4R-Gαq complex" should be changed to "H4R-Gq complex".

9. Line 111. How was rmsd calculated here and elsewhere in the manuscript? My calculations produced different results.

10. Line 133. Q347 forms weak polar interactions with the imidazole group of histamine, but not of imetit, explaining the difference in their different effects on the Q347A mutant.

11. Lines 141-142. Not clear what receptor is meant here. It doesn't apply to H4R.

12. Lines 151-153. The sentence is unclear. Why Asp94? Please, rephrase.

13. Line 159. Rephrase "which, like Trp348^{7.43} in Asp94^{3.32}".

14. Lines 161-162. Why His could not participate in hydrogen bonding with Glu?

15. Line 170. The subpocket is not really completely hydrophobic. It is formed by mostly aromatic and hydrophilic residues.

16. Line 172-176. The whole isothiurea group is likely protonated and charged. Both nitrogen atoms are equivalent. The charging state of histamine vs imetit should be described and discussed as it is important for binding of these ligands.

17. Lines 176-179. Mutant Y319A showed 5.7 fold decrease in signaling, while mutants F344A and W348A had even greater effect.

18. Lines 182-184. It would be important to note here that the imidazole ring of imetit has shifted significantly ($\sim 2 \text{ \AA}$) compared to the imidazole ring of histamine. W316 changed its rotomer, if it is real (see comment 3).

19. Line 196. Instead of “hydrophobic slot” it would be better to name this an “aromatic slot” or “aromatic subpocket” to reflect its high affinity for the cationic isothiurea group.

20. Line 199. Change to “generated by the C β –C γ bond of Phe3447.39 that rotated approximately 90 degrees”.

21. Line 276. Mutants F344I and W348A also showed a loss of histamine signaling, therefore this experiment doesn't prove that these residues are specifically responsible for imetit selectivity.

22. Line 279-284 and Extended Data Figure 5. Discussion about the “hydrophobic slot” in other aminergic receptors is confusing. None of these receptors except H4R contains Phe7.39, which was said to be important for the formation and function of the “hydrophobic slot”. Therefore this feature does not really exist in other receptors?

23. Lines 353-356. This description is confusing. Gln347 interacts with the imidazole ring of histamine. It is too far from the amine group (5 \AA). The methyl group in N α -methylhistamine is located even further and would not be affected by this residue. If the authors would like to discuss selectivity between H3R and H4R they should support their hypotheses by functional data with mutants in which specific H4R residues are mutated to H3R residues and vice versa.

24. Fig 1. How the receptor was oriented and the membrane boundaries were drawn in panels (b) and (e)? Receptor orientation does not seem to agree with that predicted by the OPM database (orientation of proteins in membranes) for H1R (too much tilt to the right).

25. Fig 2c. Inappropriate fits for Y95A, C98A, E182A, Y319A. Also Y95A in Fig. 2f. They were likely obtained by fixing the top parameter to the maximal value of the signal at the highest ligand concentration. This constraint should be removed.

26. Fig 4e. Add units for pEC50 and Emax.

27. Extended Data Fig. 1c,d. Add a scale bar.

28. Extended Data Fig. 3. Provide the number of measurements.

29. Extended Data Fig. 8. Check spelling for histamine (“histmaine” in panels b,c,d,e).

30. Supplementary Data Table 1. Add mutants Q347G and Q347L in the table. Provide the number of measurements. Units for Emax. Surface expression data are needed for all mutants to demonstrate that the absence of signal is not due to a low surface expression.

31. Cholesterol and palmitic acid were modeled in both structures but not described in the manuscript. Looking at the density, I could probably agree with cholesterol, but palmitic acid is very questionable.

32. Reporting Summary. Statistics. The exact sample size is not always reported. Statistical tests have not been performed. Null hypotheses testing has not been done. Software and code. Provide versions for all programs used.

Reviewer #2 (Remarks to the Author):

please see uploaded report

In the present paper of Iwata and coworkers the field of structural biology of histamine H4 receptors is finally opened. The H4 receptor is the latest member of the histamine receptor family, discovered only after the elucidation of the human genome. With its predominant expression in cells of the immune system, it has attracted considerable interest as new drug target. The elucidation of its structure by cryo-EM is therefore a great accomplishment and is a great step forward. In the present paper, the authors report on the H4R-Gq complexes with histamine and a related H3/H4 agonist imetit. This work is important, but the pharmacological analysis of the various mutant receptors is in my opinion not appropriate at this stage. Details on this and other issues to be tackled can be found below.

Major points:

1: At the end of the intro (line 81), the authors state that their work ‘provide insights that may contribute to the design of anti-inflammatory drugs with improved receptor selectivity’. They latter on seem to suggest that it is hard to get selectivity over H3R, but that is certainly not true, as many non-imidazole ligands have been developed as specific tools (and even approved drug in one case) for either H3R or H4R. The authors might want to be more careful in their statements.

2: (a) The authors were not able to obtain a cryo-EM structure of an H4R-Gi complex, despite Gi proteins being the most relevant G-protein partners in signaling. In contrast, it was possible to obtain a structure with a modified Gq protein (line 90). Could the authors first of all be more clear in the results section and legends, what the modified Gq protein entails? I take it that is is the mini-G approach that the authors have used? (b) Secondly, can the authors discuss in more detail the value of the current structures? What does the discussion on e.g. activation mechanism mean if one looks at a structure with an “unnatural” G-protein partner? (c) Also, do the authors have an idea why a Gi complex cannot be resolved? (d) For the actual functional assay of receptor activity the authors use a Gi-based coupling to a Gq/i1 chimeric protein. They state “a Gi-coupled H4R, but induces a Gq-dependent response”. Why did the authors not looked at an actual H4R – Gq coupling in the signaling assay in order to be able to link this to the H4R-Gq structural data?

3: line 107 states that some loops could not be observed. Please be more specific and name the specific loops that could not be modeled. Also, how about the N-terminus?

4: the authors report on the binding sites of histamine and imetit. For verification of the models for both agonists the use mutant receptors. Some of the key mutants (e.g. D94, E182) have been published already before, but references are not always mentioned at the proper spots. More importantly, the analysis the various H4R mutants that have not been analysed before are very troublesome in my opinion and need substantial more attention, especially as the results are used as “proof” for the modeled binding site. All mutants are only tested in a functional TGF- α shedding assay. In principle an appropriate assay, but any change versus the WT H4R is now presented as proof for the produced models of the binding sites. Yet, for none of the mutant is there evidence of actual expression of the mutant receptor on the cell surface nor a quantification of the expression. Whereas in this era of structural biology the pressure on new structures is high, it is imo not a good reason to forget basic molecular pharmacological rules. After cloning GPCR genes and the years of

GPCR mutagenesis studies to delineate GPCR structure-function relationships, we have learned that mutational studies can be seriously flawed by mutational effects on functional GPCR expression. And GPCR agonists will give different EC50 values for GPCRs expressed at different expression levels. None of this is controlled in the present studies. So, if in fig. 2C and F agonist do not show responses with a W348A mutant, is this because this residue is key for the binding of the ligand, or is simply the H4R protein not expressed properly in the cell membrane. Here, W348 is simple taken as example, but this holds true for many of mutants presented. So, I cannot see fig 2C as proof for “hydrogen bonding between Asp94 and Trp348”, as mentioned in line 130. I really suggest the authors to rethink the pharmacological characterization of their mutant H4Rs, as for now most of the statements that are based on the data in fig 2 are potentially flawed.

5: line 175: the authors claim that in their model the imetit binding site is formed with e.g. Tyr319 etc...Yet, in their fig 2f the mutation of Tyr319 does hardly affect the activation by imetit??

6: the binding site of both agonists is compared in detail (lines 186-214), but interestingly enough nothing is said about residue 6.48. From fig 3A one can see that W6.48 has a completely different positioning in the 2 structures. As this residue has before been pitched a being an important residue for agonist activation of family A GPCRs, it is strange that nothing is said about this residue in this section. Why would this important residue be positioned differently? The same residue is mentioned later on in the section on activation mechanism (line 304), but in this section nothing is mentioned about the differential position in the 2 agonist-bound structures.

7: The authors compare the binding sites of H4R with H1R and analyse the results of various mutants in fig 4. All the analysis (216-293, quite elaborate!) is similarly flawed as at point 4 (see above).

8: line 304: the toggle switch is discussed in relation to the imetit structure, but not with respect to histamine activation and in the HA-bound structure the W6.48 is also positioned differently. How can this be explained?

9: finally, in the section on activation mechanism the authors discuss the structure in relation to the well-known constitutive activity of H4R (line 316-338). I suggest that the authors remove this part on Gln347^{7,42}, as there is a lot of speculation and ultimately in extended data figure 7 the authors do not measure actual constitutive H4R signaling, but report on mutants that are again only characterized by histamine induced signaling. These experiments are again difficult to interpret (see point 4) and do not show any constitutive signaling of H4R unfortunately. I suggest to remove

10: in the relatively short discussion the authors mention a potential binding site for the antagonist JNJ, but no proof is given. Instead of the incomplete data focusing on constitutive activity, I suggest that the authors provide docking results of JNJ in their cryo-EM structure.

Minor points:

1: The authors state on line 51 that the “H1R was the first receptor to be discovered, followed by the H2R” and they refer to papers that deal with the cloning of their cDNA’s....I would like to highlight that in 1972 a Nobel-prize was awarded for the discovery of H2R antagonists, following the pharmacological discovery of H1R and H2R. I suggest the authors to rephrase and use appropriate references of the early work.

2: In line 54, a similar statement is used for H3R, which was already known since the 80’s of last century following a seminal paper in Nature. Also, the authors state (line 55) “it is to be expected to be a potential target”, without in fact referring to clinical use of the H3R antagonist pitolisant for narcolepsy (Wakix®) and obstructive sleep apnea (Ozawade®).

3: language needs some correction at some parts. E.g. line 89/90 does not read well. There are more examples of language issues (e.g. lines 212-214)

4: please provide the original references for the use of Nb35 and scFv16 to obtain the cryoEM structure of a GPCR-G-protein complex.

5: please label relevant residues in extended fig. 2 and other figures where this is relevant for the reader to easily get the message.

6: line 125: “ethaneamine” better be replaced by “ethylamine”.

7: line 240: the 1.6-fold increase in EC50 is statistically significant?

8: line 278/279: these lines are very difficult to understand. Please rewrite.

9: at several positions: N^α-methylhistamine.....Please use alpha as subscript

10: extended data fig 8b-e: histamine is misspelled.

11: line 563: what is a “renovated peptide”?

Reviewer #3 (Remarks to the Author):

The authors report the cryo-EM structures of the H4R-Gq complex bound with histamine and imetit, a selective H4 agonist. It is the first demonstration of such structures at high resolution for H3/4 subtypes, a Gi-coupled subfamily within H1-4 receptors. H4 has been studied for its therapeutic implications in chronic inflammatory diseases. Therefore, the novel findings by the authors add greatly to the field. Together with novel structural determination, as highlighted below, the authors identified potentially significant points (e.g., H1 vs. H4 subtype difference, hydrophobic slot, and high constitutive activity of H4). However, their findings are overcast by a lack of rigor especially in functional characterizations (written below). These points should be addressed before considering the manuscript suitable for further discussion.

Major points

- Novel H4 structures with two different agonists
- Overall congruency of key class A activation motifs (Figure 5)
- Cross-validation of ligand-pocket interaction by mutational analysis
- H1 vs. H4 subtype difference for histamine affinity/potency
- Novel “hydrophobic slot” concept
- High constitutive activity of H4 (Gln3477.42)

Critique

- The manuscript contains grammatical errors throughout that need to be corrected.
- Gq-coupling status of H4 is only supported by one citation in which chimeric G alpha proteins are used (thus not native coupling). As raised by the authors, Gi is considered as H4’s canonical G protein (many citations can be found on it.) This discrepancy needs to be acknowledged. Perhaps, some of the “negative” data of unsuccessful H4-Gi complex should be described/elaborated in the discussion rather than no data shown.
- No mentioning of G protein structure causes uninvited speculations of flaws in the study design. It should be clarified. For instance, if the Gq complex is there only to stabilize the active H4 structure. That should be stated. If the structural aspects of Gq complex are not mentioned in the manuscript because it’s not its canonical G protein, then that needs to be stated.
- Modifications on Gq need to be clearly stated (e.g., truncation of alpha-helical domain).
- Which species is used for each construct (i.e., receptor, G protein, etc)?

- bRIL and NanoBiT insertion in the structure before refinement is not immediately clear (extended figure 1). The processing point to omit these insertions should be included somewhere in the manuscript (ideally in extended figure 1).
- TGF α shedding assay with chimeric Gq/i is used as the only functional assay to validate. More G protein coupling or functional assays without mutational perturbation should be used to corroborate the TGF α shedding assay.
- Although H1 mimic of H4 by swapping residues was used in the study, H4 mimic from H1 side was not done to address histamine activity. This should be considered to further corroborate histamine activity differences.
- As above, introduction of swap-residue mutations to address isothioureia accommodation (of imetit) within H1R should be considered to complement H4-like activity in H1.
- Q347G mutation should be considered to address H4-to-H1 change in hydrophobic slot.
- Functional assays should be performed on JNJ-77

Reviewer #4 (Remarks to the Author):

Dohyun Im et al. describe the cryo-EM structures of the histamine H4 receptor coupled to the miniGsq(i) chimera and bound to histamine and imetit, a promiscuous and selective agonist respectively. The authors combine the structural information with functional TGF-shedding assays in cells to describe a change in conformation in an orthosteric Phe residue which modifies the pocket and allows for the imetit selective binding. The data provide insights into ligand selectivity, histamine receptor subtype selectivity as well as molecular mechanisms into the H4R high constitutive activity. The manuscript is well organized and clear and the insights are clearly relevant for the field. I have some concerns that the authors should address:

1.- Both EC50 and Emax of the TGF shedding assays are compared in the manuscript, however I was not able to see any experimental determination of expression levels of the different mutants that could confound the efficacy data. The authors should state whether the functional data has already been normalized by expression level of mutants, or explain why in this case it is not required.

2.- The authors do not describe the coupling of the receptor to the Gq protein. I can see how the MiniGsqi chimera could allow for non-physiological interactions, specially at the ICL2 region, however the authors should describe and compare the coupling of the H4R interface to the Gq protein, always within the limitations of the Gq chimera.

3.- The authors should show validation on the coordinate refinement such as the FSCwork/FSCtest to show the absence of modelling overfitting.

REVIEWER COMMENTS

Below we provide our replies to the reviewers' comments in blue, following the reviewers' remarks in black.

Reviewer #1 (Remarks to the Author):

This manuscript presents two cryo-EM structures of the human histamine H4 receptor bound to the endogenous agonist histamine and H3/H4 subtype selective agonist imetit in complex with engineered Gq protein, stabilized in detergent micelles by a NanoBiT tethering technology, a single chain antibody fragment scFv16, and a nanobody Nb35. The structures along with site-directed mutagenesis coupled to functional assays provided novel insights into the agonist binding mode, subtype selectivity and activation of histamine receptors. Previously, structures of the inactive and active states of the histamine H1 receptor were published. This work uncovers the first structure of the H3/H4 subtypes of the histamine receptor, which have low homology with H1/H2 subtypes, and therefore should provide considerable interest for researchers studying signaling mechanisms of the histamine receptors and for structure-based discovery of subtype selective tool compounds and drug candidates.

The structures appear to be of reasonably good quality and the study has been well executed, however, the manuscript suffers from multiple grammar issues making it difficult to read and understand. Below I provide few major and several minor critiques, addressing which should help with improving this manuscript.

We thank you for your careful review of our manuscript. We have addressed each of your comments below.

Major comments

1. It is mentioned in the manuscript that H4 receptor couples primarily to Gi protein, however, the H4R-G_i complex was unstable, and a complex with an engineered Gq protein was used instead to determine the structure. Do you have any explanations why G_q worked better than G_i? Have you tried G₁₆, which shows stronger signaling than G_q in Ref. 21. Please add units for EC₅₀ and E_{max} values in Line 91 and also include corresponding data for G_i signaling for comparison.

Thank you for pointing this out. Initially, we attempted to form a complex between H₄R and G_i and determined its expression and purity. However, based on the results of size exclusion chromatography (SEC), all peaks were skewed to the void side, indicating that these components were not forming a complex. SDS-PAGE results also showed the bands for each expressed component, but H₄R could not be confirmed, suggesting that the receptor was not stably maintained. These details will be added to Extended Data Fig. 1. It is unclear why the G_i complex could not be formed. We have not yet attempted to study G₁₆, but we plan to try forming a complex with G₁₆ in the future. We will also add the units for E_{max} in lines 91–92. Please note that pEC₅₀ is the logarithmic value of EC₅₀; thus, it does not have any units. The value corresponding to the G_i signal will also be included.

2. Related to the first question, do you consider the obtained structure to be a genuine structure of a H₄R-G_q complex or is it just an artificially constructed complex helping to obtain a structure of H₄R bound to agonists? If it's a genuine complex, then the interactions between H₄R and G_q should be described and compared to interactions between H₁R and G_q. Do these structures provide any clues why H₁R and H₂R predominantly couple to G_q while H₃R and H₄R to G_i?

Thank you for your suggestion. We believe it is possible that the H₄R-G_q complex could be formed *in vivo*. However, in this study, G_q was used as an accessory protein for structure determination, and we employed a tethering method to create an artificial complex. This does not negate the validity of the H₄R structure discussed in our study as an explanation for the binding mechanism of the agonist. Specifically, since the present study does not address the interaction between H₄R and G protein but instead focuses on the analysis of the binding mode of the ligand, we believe the discussion based on the obtained three-dimensional structure is appropriate. As previously mentioned, due to the use of an artificial H₄R-G_q complex in this study, it is difficult to discuss G protein selectivity at the histamine receptor. Recent reports have suggested that the electrostatic interaction between the receptor and G protein may be one of the important driving forces in the binding of GPCRs to G_i^{1,2}. The structures of H₄R-G_q and H₁R-G_q complexes also utilized G_{qi}N chimeric proteins (with some differences in composition). In both structures, G_α-H5 had a strong positive charge, while the receptor side that coupled to it was negatively charged. However, some

surface electrostatics of H₄R (TM2, 3, 7 on the cytoplasmic side terminus) may potentially serve as a driving force for primary G_i-coupling, with a stronger negative charge than H₁R.

1) Xu et al., *Molecular Cell*, 2021, 81, 1147–1159

2) Suno et al., *Cell Reports*, 2022, 40, 111323

3. The toggle switch residue W316^{6.48} has different conformations in structures with imetit and histamine. Are these different conformations supported by different density? And if so, do they have any implications in the signaling mechanism?

We agree with your opinion. We have also been attentive to the orientation of Trp316^{6.48} in the current structure. As advised, the orientation of the Trp316^{6.48} side chain in the current structure differs between H₄R_{ime} and H₄R_{his}, and the density map of Trp316^{6.48} in H₄R_{his} is relatively weak. Thus, we have chosen the current conformation. We further examined the model of H₄R_{his} as well as the cryo-EM density and arrived at the final conformation through structure refinement. This structure confirms that Trp316^{6.48} remains the same between H₄R_{his} and H₄R_{ime}. We will re-deposit this new structure to the PDB database. Consequently, we will redraw the relevant figures using the new structure. On the other hand, since the conformation of Trp316^{6.48} was the same in both structures is the same, it is considered that there is no effect in the signaling mechanism.

4. Signaling data with histamine (Lines 332-335) do not really support the hypothesis about the basal activity. Additionally, the Q347A mutant increased signaling potency of imetit, which is not discussed in the manuscript. What should be compared here is the basal activity of these mutants with respect to the basal activity of the wild type receptor.

We appreciate your suggestion. We concur that the signal data utilized in this study do not support the basal activity hypothesis we proposed. Therefore, we have removed the relevant content from the manuscript. The side chain stretch-out coordination of Gln347^{7,42} may potentially influence the activation mechanism of H₄R, as it is thought to control the conformation of the toggle switch, Trp316^{6,48}. Therefore, we have investigated the activation mechanism of the toggle switch by comparing it with the inactive H₃R structure and modified the manuscript to include content pertaining to this (Lines 333–349). On the other hand, the Q347A^{7,42} mutant slightly enhances the activity of imetit, but the reason for this remains unknown. Similar results were obtained in the H₁R mimic Gly and H₃R mimic Leu mutants. One possibility is that the removal of the conformation restriction on Trp316^{6,48} allows for increased interaction with the imidazole ring of imetit through closer proximity.

5. The Discussion section contains many speculations about H₃R vs H₄R selectivity that are not supported by any experimental or computer modeling data. This question is indeed interesting and important, and the author should design some specific mutations as well as try docking in H₃R model vs H₄R structure.

Thank you for your suggestion. We have performed docking simulations on the structure of H₄R obtained in this study and included them in Extended Data Fig. 9. The selective agonist OUP-16 and the selective antagonist JNJ-7777120 were found to potentially bind to the receptor by occupying a sub-pocket formed between TM5 and 6 of H₄R. This suggests that in H₃R, Tyr167^{4,57}, Thr375^{6,52}, and Met378^{6,55} hinder the formation of this sub-pocket, which may explain their specificity for H₄R. We have revised the “Discussion” section to reflect this (Lines 359–387).

Minor comments

6. Line 89. Check grammar. “The complex was able to obtain...”.

We have reviewed and revised the grammar of relevant sections in the manuscript (Lines 89–90).

7. Line 95. Incorrectly stated that resolution reflects the model to map correlation. Here the model was not used; instead, stated resolution corresponds to FCS=0.14 between two half maps.

Thank you for bringing this to our attention. As you have noted, the current representation is incorrect. The method for calculating the final resolution is described in the "Methods – Cryo-EM data processing" section; therefore, we will remove it from here (Line 95).

8. Line 100. “H₄R-G_{αq} complex” should be changed to “H₄R-G_q complex”.

We have altered all relevant names to “H₄R-G_q complex” throughout the manuscript.

9. Line 111. How was rmsd calculated here and elsewhere in the manuscript? My calculations produced different results.

We calculated the RMSD using ChimeraX – Matchmaker. When we recalculated using the newly refined models, different results were obtained. This is likely due to the use of intermediate structures. We have appended the newly calculated results to the manuscript (Lines 111–112).

10. Line 133. Q347 forms weak polar interactions with the imidazole group of histamine, but not of imetit, explaining the difference in their different effects on the Q347A mutant.

Thank you for your insight. As you pointed out, the interaction between Gln347^{7.42} and the imidazole group differs between histamine and imetit. We have reflected this information in the revised manuscript (Lines 134–135 and 186–188).

11. Lines 141-142. Not clear what receptor is meant here. It doesn't apply to H₄R.

Here, we emphasize the importance of the hydrophobic interaction between the agonist and Tyr319^{6.51} on the H₄R. It is known that the 6.51 position, along with Trp^{6.48} and Phe^{6.52}, forms an aromatic cluster in typical aminergic receptors¹. However, in H₄R, the 6.52 residue is a serine, not a phenylalanine; therefore, strictly speaking, it cannot form an aromatic cluster with these “three” residues. Nevertheless, we believe that Tyr^{6.51} and Trp^{6.48} play a crucial role in the aromatic stacking between the agonist and receptor; therefore, the corresponding expression should not be problematic.

1) Kooistra et al., *Br. J. Pharmacol*, 2013, 170, 101–126

12. Lines 151-153. The sentence is unclear. Why Asp94? Please, rephrase.

Thank you for pointing this out. There was an inaccuracy in word choice. We have corrected the relevant sentence (Lines 153–155).

13. Line 159. Rephrase “which, like Trp348^{7.43} in Asp94^{3.32}”.

We have rewritten the corresponding text of the manuscript (Lines 160–163).

14. Lines 161-162. Why His could not participate in hydrogen bonding with Glu?

Thank you for your comment. As you pointed out, His^{4.56} may form hydrogen bonds with Glu^{5.46}. Unfortunately, we misinterpreted the intent of the reference literature. It has been suggested that the bulky side chain of His^{4.56} causes a shift in the coordination of Glu^{5.46} to TM5 and 6, resulting in the reformation of the pocket and weakening of agonist binding. In other words, Asn^{4.56} is thought to optimally regulate agonist binding by delicately shaping the pocket through interaction with Glu^{5.46}. Therefore, this position is considered an important residue in explaining species-specific differences in histamine affinity. We have modified the relevant text in the manuscript (Lines 163–165).

15. Line 170. The subpocket is not really completely hydrophobic. It is formed by mostly aromatic and hydrophilic residues.

Thank you for your suggestion. As you correctly pointed out, this pocket is formed by three hydrophobic/aromatic residues and Gln. We have rewritten the text to correct this (Lines 172–174).

16. Line 172-176. The whole isothiourea group is likely protonated and charged. Both nitrogen atoms are equivalent. The charging state of histamine vs imetit should be described and discussed as it is important for binding of these ligands.

Thank you for your suggestion regarding the charging state of the ligands. We have added a discussion on the charging state of ligands and its effect on binding to the manuscript (Lines 204–210).

17. Lines 176-179. Mutant Y319A showed 5.7 fold decrease in signaling, while mutants F344A and W348A had even greater effect.

We have included details about the differences in activity between mutants in the manuscript (Lines 179–181).

18. Lines 182-184. It would be important to note here that the imidazole ring of imetit has shifted significantly (~2 Å) compared to the imidazole ring of histamine. W316 changed its rotamer, if it is real (see comment 3).

As previously noted in Comment 3, the rotamer of Trp316^{6,48} in H₄R_{his} was incomplete. We have since refined the structure and confirmed that there is no difference between H₄R_{his} and H₄R_{ime}. This suggests that there is no correlation between the conformation of Trp316^{6,48} and the coordination of the imidazole ring of each agonist.

19. Line 196. Instead of “hydrophobic slot” it would be better to name this an “aromatic slot” or “aromatic subpocket” to reflect its high affinity for the cationic isothiourea group.

Thank you for your suggestion. We have changed the name of this sub-pocket to “aromatic slot”, according to your suggestion.

20. Line 199. Change to “generated by the C β –C γ bond of Phe344^{7.39} that rotated approximately 90 degrees”.

The relevant passage in the manuscript has been amended (Lines 202–204).

21. Line 276. Mutants F344I and W348A also showed a loss of histamine signaling, therefore this experiment doesn’t prove that these residues are specifically responsible for imetit selectivity.

Thank you for your point. As you mentioned, the F344I^{7.39} and W348Y^{7.43} mutants reduce not only the signaling activity of imetit but also histamine activity; therefore, the discussion on subtype selectivity in this mutant experiment may not be sufficient. However, considering that the binding affinity of histamine is more than 100-fold lower in H₁R than that in H₄R, the fact that the introduction of only one mutation of the H₁R type prevents the detection of histamine signaling activity may be consistent.

22. Line 279-284 and Extended Data Figure 5. Discussion about the “hydrophobic slot” in other aminergic receptors is confusing. None of these receptors except H₄R contains Phe^{7.39}, which was said to be important for the formation and function of the “hydrophobic slot”. Therefore this feature does not really exist in other receptors?

Thank you for your comment. As you have pointed out, the current discussion may be confusing. Here, we simply demonstrate that the “hydrophobic slot” observed in H₄R possesses a highly unique topology throughout the entire aminergic receptor. Smartly utilizing this sub-pocket may provide useful information for designing novel compounds specialized for H₄R (or H₃R). Therefore, we have revised the manuscript to reflect this (Lines 299–302).

23. Lines 353-356. This description is confusing. Gln347 interacts with the imidazole ring of histamine. It is too far from the amine group (5 Å). The methyl group in N α -methylhistamine is located even further and would not be affected by this residue. If the authors would like to discuss selectivity between H₃R and H₄R they should support their hypotheses by functional data with mutants in which specific H₄R residues are mutated to H₃R residues and vice versa.

We concur with the reviewer and have determined that using 4-methylhistamine would make it difficult to support our claims. As previously addressed in Comment 5, we have conducted docking simulations with OUP-16 and JNJ-7777120 to discuss the selectivity between H₃R and H₄R and have included this information in the “Discussion” section.

24. Fig 1. How the receptor was oriented and the membrane boundaries were drawn in panels (b) and (e)? Receptor orientation does not seem to agree with that predicted by the OPM database (orientation of proteins in membranes) for H₁R (too much tilt to the right).

We have redrawn the figure with the correct orientation of H₄R in the membrane according to the orientation of H₁R in the OPM database, as you suggested (Fig. 1).

25. Fig 2c. Inappropriate fits for Y95A, C98A, E182A, Y319A. Also Y95A in Fig. 2f. They were likely obtained by fixing the top parameter to the maximal value of the signal at the highest ligand concentration. This constraint should be removed.

We have reproduced the optimal fitting curve upon re-evaluating the fitting method for these mutants (Fig. 2c, f, Extended Data Fig. 4b).

26. Fig 4e. Add units for pEC₅₀ and Emax.

Thank you for your proposal. pEC₅₀ refers to negative values of logarithmically transformed EC₅₀ values and thus does not have a unit. The Emax value used in this manuscript is derived from the results of a TGF α shedding assay, specifically the top values of percent AP-TGF α release, and thus its unit has been added to the figure (Fig. 4e).

27. Extended Data Fig. 1c,d. Add a scale bar.

We have added a scale bar to the figure (Extended Data Fig. 2c, d).

28. Extended Data Fig. 3. Provide the number of measurements.

Thank you for your suggestion. We have added the number of measurements in the figure legend (Extended Data Fig. 4).

29. Extended Data Fig. 8. Check spelling for histamine (“histmaine” in panels b,c,d,e).

Thank you for pointing this out. We have corrected the typo in the figure (Extended Data Fig. 9).

30. Supplementary Data Table 1. Add mutants Q347G and Q347L in the table. Provide the number of measurements. Units for E_{max}. Surface expression data are needed for all mutants to demonstrate that the absence of signal is not due to a low surface expression.

Thank you for your suggestion. We have removed the discussion on the constitutive activity of H₄R using the relevant mutant data in the revised manuscript and removed the corresponding data. A discussion using Q347G has been added to the "Subtype-selectivity of H₄R and H₁R" section (Lines 259–265, Lines 291–297). We have also included the number of measurements taken and the units for E_{max} (Supplementary Data Table 1). We have measured surface expression for all mutants and have included the results in the manuscript (Extended Data Fig. 4).

31. Cholesterol and palmitic acid were modeled in both structures but not described in the manuscript. Looking at the density, I could probably agree with cholesterol, but palmitic acid is very questionable.

We concur with your suggestion. We have removed the model of palmitic acid and performed further refinement.

32. Reporting Summary. Statistics. The exact sample size is not always reported. Statistical tests have not been performed. Null hypotheses testing has not been done. Software and code. Provide versions for all programs used.

Thank you for your comments. We will fill out and submit all the required fields in the Reporting Summary, which has been updated to the latest version.

Reviewer #2 (Remarks to the Author):

In the present paper of Iwata and coworkers the field of structural biology of histamine H4 receptors is finally opened. The H4 receptor is the latest member of the histamine receptor family, discovered only after the elucidation of the human genome. With its predominant expression in cells of the immune system, it has attracted considerable interest as new drug target. The elucidation of its structure by cryo-EM is therefore a great accomplishment and is a great step forward. In the present paper, the authors report on the H₄R-G_q complexes with histamine and a related H₃/H₄ agonist imetit. This work is important, but the pharmacological analysis of the various mutant receptors is in my opinion not appropriate at this stage. Details on this and other issues to be tackled can be found below.

We wish to express our appreciation to the reviewer for these insightful comments, which have helped us significantly improve the paper. We have addressed each of your comments below.

Major points:

1: At the end of the intro (line 81), the authors state that their work ‘provide insights that may contribute to the design of anti-inflammatory drugs with improved receptor selectivity’. They latter on seem to suggest that it is hard to get selectivity over H₃R, but that is certainly not true, as many non-imidazole ligands have been developed as specific tools (and even approved drug in one case) for either H₃R or H₄R. The authors might want to be more careful in their statements.

Thank you for your observation. As you noted, non-imidazole ligands are utilized as specific tools for H₃R and H₄R. In addition to the structural characteristics of ligands serving as selection tools,

we believe that our structural information of target protein could aid in the rational development of H₄R-targeted therapeutics. Given that statements regarding selectivity may engender confusion, we have elected to remove them here (Lines 80–82).

2: (a) The authors were not able to obtain a cryo-EM structure of an H₄R-G_i complex, despite G_i proteins being the most relevant G-protein partners in signaling. In contrast, it was possible to obtain a structure with a modified G_q protein (line 90). Could the authors first of all be more clear in the results section and legends, what the modified G_q protein entails? I take it that is the mini-G approach that the authors have used? (b) Secondly, can the authors discuss in more detail the value of the current structures? What does the discussion on e.g. activation mechanism mean if one looks at a structure with an “unnatural” G-protein partner? (c) Also, do the authors have an idea why a G_i complex cannot be resolved? (d) For the actual functional assay of receptor activity the authors use a G_i-based coupling to a Gq/i1 chimeric protein. They state “a G_i-coupled H₄R, but induces a Gq-dependent response”. Why did the authors not look at an actual H₄R – Gq coupling in the signaling assay in order to be able to link this to the H₄R-Gq structural data?

Thank you for your valuable feedback.

(a) To prevent confusion, the word "modified" has been deleted, and G_q has been replaced by miniG_{qi}N (Lines 89–90). Details on miniG_{qi}N are described in the "Method – Constructs" section. Any missing explanations have been added in Lines 564–566. The G_q used in this study is a G_q sequence substitute for the $\alpha 5$ helix of the C-terminal of the mini-G_s skeleton. Two dominant negative mutants have also been introduced, and the 35 residues at the N-terminus have been transplanted with the G_i sequence to allow binding by scFv16.

(b) We think that the structure does not impede the examination of the agonist-bound attributes of H₄R. Given the presence of an atypical G-protein, the discourse regarding the receptor-G-protein interface is deemed invalid. Nevertheless, the structure of H₄R obtained in this study, when compared with other active-state receptors, is definitively in an active conformation and does not hinder discussion on the mechanism of agonist binding and receptor activation.

(c) We have appended the results of the expression and purification of H₄R-G_i to Extended Data Fig. 1. We attempted expression and purification using the tethering method in the formation of the complex with G_i, but we did not achieve complex generation. Honestly, we do not know the reason.

From the results of SEC during purification, most peaks were detected on the void side. Although the bands of each component, such as G-protein, were confirmed by SDS-PAGE, the H₄R band was unknown, suggesting that the expression of the receptor was weak or its stability was not maintained.

(d) The primary signaling partner of H₄R is G_i; however, there have been reports of a weak G_q signal¹. In the current manuscript, we measured G_i signaling (precisely the G_{q/i1} chimera protein), which is the most prioritized coupling partner of H₄R, and evaluated each variant. On the other hand, this structure is the G_q-coupled complex, and if the G_q signal can be measured, it is believed that consensus would be achieved with structure data. However, due to the weak signal intensity, we were unable to detect the signal of wild-type G_q in our experimental system. Nevertheless, this study mainly involves experiments on the mutations of residues around the ligand binding site, and by comparing them with the WT, we have progressed the discussion and determined that there should be no issue with using the G_i signal assay data.

1) Inoue et al., *Cell*, 2019, 177 (7), 1933–1947

3: line 107 states that some loops could not be observed. Please be more specific and name the specific loops that could not be modeled. Also, how about the N-terminus?

Thank you for your suggestion. We have included a detailed description of the following regions in the manuscript, which are not modeled in the structures (Lines 105–108).

H₄R_{his}: N terminal (1–11), ECL2 (159–160), ICL3 (205–297), ECL3 (330–335), C-terminal (374–390)

H₄R_{ime}: N terminal (1–11), ECL2 (159–160), ICL3 (205–297), ECL3 (330–335), C-terminal (374–390)

4: the authors report on the binding sites of histamine and imetit. For verification of the models for both agonists the use mutant receptors. Some of the key mutants (e.g. D94, E182) have been published already before, but references are not always mentioned at the proper spots. More importantly, the analysis the various H₄R mutants that have not been analysed before are very

troublesome in my opinion and need substantial more attention, especially as the results are used as “proof” for the modeled binding site. All mutants are only tested in a functional TGF- α shedding assay. In principle an appropriate assay, but any change versus the WT H4R is now presented as proof for the produced models of the binding sites. Yet, for none of the mutant is there evidence of actual expression of the mutant receptor on the cell surface nor a quantification of the expression. Whereas in this era of structural biology the pressure on new structures is high, it is imo not a good reason to forget basic molecular pharmacological rules. After cloning GPCR genes and the years of GPCR mutagenesis studies to delineate GPCR structure-function relationships, we have learned that mutational studies can be seriously flawed by mutational effects on functional GPCR expression. And GPCR agonists will give different EC₅₀ values for GPCRs expressed at different expression levels. None of this is controlled in the present studies. So, if in fig. 2C and F agonist do not show responses with a W348A mutant, is this because this residue is key for the binding of the ligand, or is simply the H4R protein not expressed properly in the cell membrane. Here, W348 is simple taken as example, but this holds true for many of mutants presented. So, I cannot see fig 2C as proof for “hydrogen bonding between Asp94 and Trp348”, as mentioned in line 130. I really suggest the authors to rethink the pharmacological characterization of their mutant H4Rs, as for now most of the statements that are based on the data in fig 2 are potentially flawed.

Thank you very much for your meticulous and polite observations. Indeed, verifying the expression levels of each mutant in accordance with what you have suggested is deemed a fundamental principle in molecular pharmacology. I believe that after quantifying the expression, signal assays should be performed for each mutant. In response to your observations, we have confirmed the cell surface expression of all mutants. As a result, no mutants with a significantly decreased cell surface expression were found. Therefore, we have determined that the data of each mutant used in this pharmacological characterization evaluation is reliable. We have added the verification data on cell surface expression for each mutant receptor to Extended Data Fig. 4a.

5: line 175: the authors claim that in their model the imetit binding site is formed with e.g. Tyr319 etc...Yet, in their fig 2f the mutation of Tyr319 does hardly affect the activation by imetit??

Thank you for pointing that out. The isothiourea group of imetit utilizes the hydrophobic pocket comprised of Tyr319, Phe344, Gln347, and Trp348 for its binding. This isothiourea group undergoes pi-cation interaction with Tyr319 and is protonated and charged, thus forming relatively strong hydrogen bonds with nearby Asp94 and Phe344. The activity is abolished in the mutants of Asp94 and Phe344, and the reduced activity (approximately 5 times) in the mutant of Tyr319 is thought to be the reason.

6: the binding site of both agonists is compared in detail (lines 186-214), but interestingly enough nothing is said about residue 6.48. From fig 3A one can see that W6.48 has a completely different positioning in the 2 structures. As this residue has before been pitched a being an important residue for agonist activation of family A GPCRs, it is strange that nothing is said about this residue in this section. Why would this important residue be positioned differently? The same residue is mentioned later on in the section on activation mechanism (line 304), but in this section nothing is mentioned about the differential position in the 2 agonist-bound structures.

Thank you for your comment. As you pointed out, the orientation of Trp316^{6.48} is different in H4R_{ime} and H4R_{his}. The electron density of Trp316^{6.48} in H4R_{his} was slightly weak, which led to the current conformation. However, after thoroughly examining the structure and density, the refinement was performed again, and a final structure was obtained. This new structure confirms no change in Trp316^{6.48} between H4R_{ime} and H4R_{his}. We have re-deposited this new structure in the PDB database. It is believed that both histamine and imetit's imidazole ring interact with Trp316^{6.48} through stacking interactions. This residue is known as a "toggle switch," and it is thought that receptor activation proceeds when the agonist interacts with this residue in H4R. Therefore, we have included this content in the "Activation mechanism of H4R" section.

7: The authors compare the binding sites of H4R with H1R and analyse the results of various mutants in fig 4. All the analysis (216-293, quite elaborate!) is similarly flawed as at point 4 (see above).

Thank you for your observations. As previously addressed in Point 4, we have confirmed the surface expression levels of all mutant receptors and have included the results in Extended Data Fig.4a.

8: line 304: the toggle switch is discussed in relation to the imetit structure, but not with respect to histamine activation and in the HA-bound structure the W6.48 is also positioned differently. How can this be explained?

Thank you for your suggestion. Although it is similar to the response provided in Point 6, the interaction between the “toggle switch” (Trp316^{6.48}) and histamine or imetit is explained in the "Molecular basis for the recognition of histamine and imetit by H₄R" section (Lines 145–147 and 185–186). The orientation of its side chain was corrected by re-refinement of the H₄R_{his} structure. As a result, it was confirmed to be the same conformation as the structure of H₄R_{ime}. The activation mechanism of the “toggle switch” in H₄R is newly described in Lines 333–349.

9: finally, in the section on activation mechanism the authors discuss the structure in relation to the well-known constitutive activity of H₄R (line 316-338). I suggest that the authors remove this part on Gln347^{7.42}, as there is a lot of speculation and ultimately in extended data figure 7 the authors do not measure actual constitutive H₄R signaling, but report on mutants that are again only characterized by histamine induced signaling. These experiments are again difficult to interpret (see point 4) and do not show any constitutive signaling of H₄R unfortunately. I suggest to remove

Thank you for your observation. We concur that the association between Gln347^{7.42} and constitutive activity is tenuous at best. Accordingly, we have removed the relevant content from the manuscript.

10: in the relatively short discussion the authors mention a potential binding site for the antagonist JNJ, but no proof is given. Instead of the incomplete data focusing on constitutive activity, I suggest that the authors provide docking results of JNJ in their cryo-EM structure.

Thank you for your suggestion. As you mentioned, there was much speculation about the binding of JNJ-7777120. Based on your suggestion, we have performed a docking simulation on the cryo-EM structure of H₄R. The results have been added to Lines 383–387 and Extended Data Fig. 9e. Our docking results showed that the chloro-indole moiety of JNJ-7777120 can occupy the sub-pocket formed between TM5 and TM6, which is in line with our predictions. The fact that Met378^{6,55}, which has a relatively bulky and hydrophobic side chain, is present in the region corresponding to the sub-pocket and cannot form this sub-pocket in H₃R supports the H₄R-selective nature of JNJ-7777120. Of course, since JNJ-7777120 is an antagonist, the binding prediction in the active structure is only reference data. Our next challenge is to clarify the inactive structure of H₄R with bound JNJ-7777120.

Minor points:

1: The authors state on line 51 that the “H₁R was the first receptor to be discovered, followed by the H₂R” and the refer to papers that deal with the cloning of their cDNA’s....I would like to highlight that in 1972 a Nobel-prize was awarded for the discovery of H₂R antagonists, following the pharmacological discovery of H₁R and H₂R. I suggest the authors to rephrase and use appropriate references of the early work.

Thank you for your comments regarding the background of histamine receptors. Indeed, the cloning of the receptors is not the only important aspect, but also the pharmacological discoveries and their contributions to the Nobel Prize award. Therefore, we have revised the manuscript to reflect this and added appropriate references (Lines 50–52).

2: In line 54, a similar statement is used for H₃R, which was already known since the 80’s of last century following a seminal paper in Nature. Also, the authors state (line 55) “it is to be expected to be a potential target”, without in fact referring to clinical use of the H₃R antagonist pitolisant for narcolepsy (Wakix®) and obstructive sleep apnea (Ozawade®).

Thank you for your suggestion. We have changed the relevant part of the introduction regarding the discovery of H₃R. We have also added a note about the H₃R antagonist therapeutics that have been launched (Lines 53–57).

3: language needs some correction at some parts. E.g. line 89/90 does not read well. There are more examples of language issues (e.g. lines 212-214)

Thank you for pointing that out. We also think there were language issues that made it a bit confusing. We have corrected these to make reading easier (Lines 89–90, 218–220).

4: please provide the original references for the use of Nb35 and scFv16 to obtain the cryoEM structure of a GPCR-G-protein complex.

Thank you for your suggestion. We have added the correct references for Nb35¹ and scFv16².

1) Rasmussen et al., *Nature*, 2011, 19;477(7366):549–55.

2) Maeda et al., *Nat Commun*, 2018, 13;9(1):3712

5: please label relevant residues in extended fig. 2 and other figures where this is relevant for the reader to easily get the message.

Thank you for pointing that out. To make it easier for the reader to understand, appropriate labels have been added to the relevant figures (Extended Data Fig. 3, Extended Data Fig. 7).

6: line 125: “ethaneamine” better be replaced by “ethylamine”.

We have replaced the word in question with the appropriate one (Line 125).

7: line 240: the 1.6-fold increase in EC50 is statistically significant?

Thank you for your observations. We found that there was an error in the data analysis. As you mentioned, the N147Y^{4.57} of the H₃R mimic mutant showed a change in EC₅₀ of approximately 1.6 times compared with that of the WT, but it was not a “significant” change. It is considered that the Tyr at the 4.57 position interacts with Glu182^{5.46} in the same way as Asn, stabilizing its side

chain. The fact that the N147W^{4.57} of the H₁R type lost its signal activity, while the N147Y^{4.57} of the H₃R type showed no remarkable change compared with that of the WT, supports the subtype selectivity of histamine activity seen between H₁R/H₂R and H₃R/H₄R. We have incorporated this into the manuscript (Lines 244–250).

8: line 278/279: these lines are very difficult to understand. Please rewrite.

Thank you for pointing this out. The wording is indeed very difficult to understand. Therefore, this section has been rewritten (Lines 295–297).

9: at several positions: Na-methylhistamine.....Please use alpha as subscript

Thank you for your suggestion. We have determined that using 4-methylhistamine would make it difficult to support our claims. We have conducted docking simulations with OUP-16 and JNJ-7777120 to discuss the selectivity between H₃R and H₄R and have included this information in the “Discussion” section.

10: extended data fig 8b-e: histamine is misspelled.

We have corrected the misspelling of the histamine (Extended Data Fig. 9).

11: line 563: what is a “renovated peptide”?

I apologize for the lack of clarity in the manuscript. The "renovated peptide" refers to HiBiT, which is a high-affinity peptide comprised of 11 amino acids. HiBiT is one of two fragments developed based on NanoLuc[®] luciferase and was improved from Promega's NanoLuc[®] to NanoBiT[®]. To avoid confusion, the word "renovated" has been removed and replaced with a new statement to clearly convey its meaning (Lines 568–571).

Reviewer #3 (Remarks to the Author):

The authors report the cryo-EM structures of the H4R-Gq complex bound with histamine and imetit, a selective H4 agonist. It is the first demonstration of such structures at high resolution for H3/4 subtypes, a Gi-coupled subfamily within H1-4 receptors. H4 has been studied for its therapeutic implications in chronic inflammatory diseases. Therefore, the novel findings by the authors add greatly to the field. Together with novel structural determination, as highlighted below, the authors identified potentially significant points (e.g., H1 vs. H4 subtype difference, hydrophobic slot, and high constitutive activity of H4). However, their findings are overcast by a lack of rigor especially in functional characterizations (written below). These points should be addressed before considering the manuscript suitable for further discussion.

Major points

- Novel H4 structures with two different agonists
- Overall congruency of key class A activation motifs (Figure 5)
- Cross-validation of ligand-pocket interaction by mutational analysis
- H1 vs. H4 subtype difference for histamine affinity/potency
- Novel “hydrophobic slot” concept
- High constitutive activity of H4 (Gln3477.42)

We thank the reviewer for the time taken to carefully review our manuscript and for the appreciation of our manuscript and its potential impact. We have answered each of their comments below.

1. The manuscript contains grammatical errors throughout that need to be corrected.

Thank you for pointing this out. We are aware that there are numerous grammatical errors throughout the manuscript. We have corrected the erroneous parts.

2. Gq-coupling status of H4 is only supported by one citation in which chimeric G alpha proteins are used (thus not native coupling). As raised by the authors, Gi is considered as H4’s canonical G protein (many citations can be found on it.) This discrepancy needs to be acknowledged. Perhaps,

some of the “negative” data of unsuccessful H4-Gi complex should be described/elaborated in the discussion rather than no data shown.

Thank you for your observation. We concur that the standard signaling protein for H4R is G_i, and we attempted to form a complex with H4R-G_i before this study. However, we were unable to achieve the formation of this complex. In this present study, we have determined the complex structure of H4R-G_q and mainly discussed the ligand binding features of H4R. Nevertheless, we agree with your observation that presenting the "negative" data on the failed attempt to form an H4R-G_i complex is necessary. The purification profiles for the H4R-G_i complex have been attached in the revised manuscript (Extended Data Fig. 1b).

3. No mentioning of G protein structure causes uninvited speculations of flaws in the study design. It should be clarified. For instance, if the G_q complex is there only to stabilize the active H4 structure. That should be stated. If the structural aspects of G_q complex are not mentioned in the manuscript because it's not its canonical G protein, then that needs to be stated.

Thank you for your suggestion. Indeed, the G_q used in the present study was utilized to stabilize the structure of H4R, which was previously described as an "accessory protein" in the manuscript, but this description may not have been robust. As you pointed out, we have added the fact that G_q is a non-canonical G protein with respect to H4R (Lines 89–90).

4. Modifications on G_q need to be clearly stated (e.g., truncation of alpha-helical domain).

Thank you for your comment. We acknowledge that the explanation regarding the modification of G_q was insufficient. Although a description of G_q modification was presented in the “Methods – Constructs” session, we have added a further explanation of the details of the modification (Lines 564–566).

5. Which species is used for each construct (i.e., receptor, G protein, etc)?

The species of each construct used in this study are listed in the revised manuscript. These are already described in the “Methods – Constructs” session. Histamine H4 receptor: Human (Line 560), G_q: Chimera protein (Lines 564–566), G_β: Rat (Line 572), G_γ: Bovine (Line 572).

6. bRIL and NanoBiT insertion in the structure before refinement is not immediately clear (extended figure 1). The processing point to omit these insertions should be included somewhere in the manuscript (ideally in extended figure 1).

Thank you for your comment. The explanation regarding bRIL and NanoBiT insertion in data processing was insufficient. Firstly, a cleavage sequence of 3C protease is present between bRIL and the receptor, and during purification, the bRIL fraction is excluded by the 3C protease treatment. To prevent confusion regarding construct composition, we have added a snake plot of the H₄R construct used in this study to Extended Data Fig. 1a. Moreover, in the case of NanoBiT, the flexibility is higher compared with that of the receptor and G proteins, resulting in weaker density corresponding to NanoBiT on average (this is also the case for micelles). We did not intentionally exclude them using the mask; therefore, the final map contains a weak density of NanoBiT, but we did not model it.

7. TGF α shedding assay with chimeric Gq/i is used as the only functional assay to validate. More G protein coupling or functional assays without mutational perturbation should be used to corroborate the TGF α shedding assay.

Thank you for your feedback. In this study, we utilized the TGF α shedding assay as the primary functional assay to validate our hypothesis. The utilization of this assay system to evaluate signal activity has already been adopted in many other studies. Furthermore, we used this experimental system to compare the activity of the mutant receptor and the wild-type around the ligand-binding pocket; therefore, we believe that there is no issue in evaluating the validity of ligand binding. However, since we have not confirmed the expression of each mutant receptor, we have added the results of cell surface expression level evaluation by flow cytometry (Extended Data Fig. 4a).

8. Although H1 mimic of H4 by swapping residues was used in the study, H4 mimic from H1 side was not done to address histamine activity. This should be considered to further corroborate histamine activity differences.

We have responded to comments 8 and 9 together.

9. As above, introduction of swap-residue mutations to address isothiourrea accommodation (of imetit) within H1R should be considered to complement H4-like activity in H1.

Thank you for your comment. We will respond to the two comments (#8 and #9) mentioned above. Indeed, in this study, we validated our hypothesis regarding the subtype selectivity of H₄R and other histamine receptors through cryo-EM structures and mutagenesis functional assays. Basically, all discussions in this study were based on the obtained H₄R structure, and the H₁R swap mutant was also evaluated similarly. The evaluation showed that H₄R and H₁R have different recognition processes for ligands, which we believe are sufficient to support our claims. As you pointed out, the verification of H₄R mimicking the H₁R mutant (2-way verification) would strengthen the subtype selectivity that we argue in this study, but there is a slight concern that it would deviate from the main aim of the paper, which is to advance discussions based on the structure of H₄R. Therefore, we would like to continue this discussion with just the H₄R mutant in this manuscript. However, if necessary, we would like to make further changes (additional mutant creation).

10. Q347G mutation should be considered to address H4-to-H1 change in hydrophobic slot.

Thank you for your suggestion. We have conducted experiments on the H₁R/H₂R-type mutant Q347G^{7,42}. The data has been added to Figs. 4e, f, Extended Data Fig. 4, and Supplementary Table 1. The results of this mutant towards histamine and imetit have been included in the manuscript (Lines 259–265, Lines 291–297).

11. Functional assays should be performed on JNJ-77

We do consider JNJ-7777120 to be a crucial ligand for discussing the selectivity of H₃R and H₄R and have carried out docking simulations on the H₄R model to better complement our arguments. These simulations have been added to the revised version (Lines 383–387, Extended Data Fig. 9). Given that JNJ-7777120 is an antagonist of H₄R, it would be challenging to include it in the functional assay adopted in this study. Our aim in future studies is to clarify the structure and function of inactive H₄R. We anticipate that our future studies will provide clear evidence on this subject.

Reviewer #4 (Remarks to the Author):

Dohyun Im et al. describe the cryo-EM structures of the histamine H₄ receptor coupled to the miniGsq(i) chimera and bound to histamine and imetit, a promiscuous and selective agonist respectively. The authors combine the structural information with functional TGF-shedding assays in cells to describe a change in conformation in an orthosteric Phe residue which modifies the pocket and allows for the imetit selective binding. The data provide insights into ligand selectivity, histamine receptor subtype selectivity as well as molecular mechanisms into the H₄R high constitutive activity. The manuscript is well organized and clear and the insights are clearly relevant for the field. I have some concerns that the authors should address:

Thank you for your appreciative review and evaluation of the potential of this paper. We have responded to each of your comments below.

1. Both EC₅₀ and E_{max} of the TGF shedding assays are compared in the manuscript, however I was not able to see any experimental determination of expression levels of the different mutants that could confound the efficacy data. The authors should state whether the functional data has already been normalized by expression level of mutants, or explain why in this case it is not required.

Thank you for your comment. We agree that verifying the expression levels of each mutant is necessary to discuss the pharmacological data in this study. We have confirmed the cell surface expression of all mutants and found no significant impact on their expression. The data have been added to Extended Data Fig. 4a.

2. The authors do not describe the coupling of the receptor to the Gq protein. I can see how the MiniGsqi chimera could allow for non-physiological interactions, specially at the ICL2 region, however the authors should describe and compare the coupling of the H4R interface to the Gq protein, always within the limitations of the Gq chimera.

Thank you for your valuable feedback. The H₄R is supposed to primarily couple with G_i. However, we were unable to obtain the complex in this study. Thus, we employed miniGq_iN as an accessory protein to stabilize the H₄R, thereby elucidating the structure of H₄R. Consequently, we believe that a discussion on the interface between H₄R and G_q is outside of the scope of this work. As you pointed out, the G_q chimera exhibits non-physiological interactions with the ICL2 of H₄R; however, we would prefer to avoid discussing it in detail within the context of this paper. Instead, we added negative data, reflecting that we could not form a complex with G_i, to Extended Data Fig. 1.

3.- The authors should show validation on the coordinate refinement such as the FSC_{work}/FSC_{test} to show the absence of modelling overfitting.

Thank you for your suggestion. We agree that validating the modeling is necessary. Therefore, we have performed cross-validation of the coordinate refinement and added the FSC_{work}/FSC_{test} to the revised manuscript (Extended Data Fig. 2).

REVIEWER COMMENTS

Reviewer #1 (Remarks to the Author):

The authors addressed my questions, and the manuscript has been greatly improved. I can now support its publication in Nature Communications.

Reviewer #2 (Remarks to the Author):

In the present revision Im et al. present an updated version of their earlier work on H4R-agonist cryo-EM studies. It is good to see that a fair number of the comments have been addressed and that e.g. the position of the toggle switch has now been corrected, details on constructs and “undetermined parts of the structures” are now available.

Yet, also in the present version still a number of details on methodology/statistics etc are lacking. Moreover, the MS has been seriously rewritten, which at some parts raises again some issues. Also, a number of points have not been properly addressed and need imo further attention.

Major points:

1: Introduction has been improved with references to the earlier work of Black/Arrang etc. Yet, the statement on the therapeutic use of pitolisant needs at least one reference. Also, it seems to me that the statements in line 80-82 have not been removed, as mentioned in the rebuttal?

2: a) it is nice to see the details on the miniG protein construct used.

b) the authors also claim that the structure with an unnatural G-protein is useful (despite the fact that Gq can hardly be activated by H4R in the hands of the authors themselves as indicated at 2d), by stating that “the protein is definitely in an active conformation”, without providing the details why they state this. I think the reader needs a bit of help to understand the stat value of the present complexes and that such a statement should be carefully discussed in the discussion of the MS.

c) it is nice to see the extra data in extended fig 1, but no methods can be found for the experiments in fig. 1B for the H4R-Gi complex. Also, from the gel one can deduce that there is also another difference between the 2 efforts...the use of Nb35. In the paper it would be good to explicitly mention where the Nb35 is binding to the complex (and also for the svFv16).

Did the authors try the same approach as with “the artificial Gq, that was based on the miniGs scaffold etc”? Did the authors try an artificial Gi, based on the same approach??

d) the authors should state clearly in the MS that they have been unable to measure Gq signaling with the WT H4R and discuss this in the context of their H4R complexes.

3: good to see the details mentioned now

4: Still open comment: earlier paper(s) that have already been analyzing some of the H4R mutants in this paper are not mentioned. I do not appreciate this attitude. Proper referencing is good academic ethics.

With respect to the analysis of the mutants, the authors made some efforts, but I still have issues with the use of these data to support the cryo-EM data....The methods used to detect the expression are lacking (like for a fair number of other additions). What was used to detect functional expression? I presume some kind of antibody?? But which one and how, especially as there has been a discussion in the field on the use of commercial H4R antibodies? Also, the authors state “no mutants with significantly decreased cell surface expression were found”. Looking at the new figs Ext data 4a, I do not see any statistical analysis. The authors should add a statistical analysis to the figure and to the methods....and to me it seems that a fair number of mutants have a significantly decreased expression (even clear without statistical analysis) and I do not understand why the authors use the statement, mentioned above.

5: although fig. 2f does not show a huge effect of the Tyr319Ala mutation, the authors report that the EC50 is changed approximately 5-fold. First of all, there are other changes that are \pm 5-fold (e.g. Q349G). Does that also highlight an important interaction?

More importantly, there is no statistical analysis of the data in the Supp data Table 1. The authors should add such an analysis for all mutants and then discuss the changes observed. Also, would a lower expression of the mutant explain the 5-fold shift perhaps?

6: that authors corrected the position of the important 6.48 residue.

7: see point 2 on the mutations

8: see 6

9: the authors followed the earlier suggestion.

10: JNJ777120 docking in the active structure has now been added, which I suggested in the first round of review instead of the somewhat artificial focus on constitutive activity (and that was not shown experimentally). As JNJ is potentially a biased agonist, this approach is perhaps not as uncommon as it may seem, but the present work is not positioned in the present paper like this. Now it is really odd to read that an antagonist is being docked into an active structure. I also suggest that the authors do MD studies to allow the system to equilibrate to a potentially inactive structure? Also, how do the present results compare to docking in e.g. an inactive H4R model present at GPCRDb and the AlphaFold model? Moreover, there are some mutagenesis data in the literature on JNJ binding and the current pose should be considered in the context of such data.

A minor point: methods of the computational approach cannot be found and the legend of the figure is not mentioning the JNJ docking pose.

Extra: Abstract need some editing (histamine agonist do not exist...also, one can not talk about “subtype selectivity of histamine receptors”, although I understand what the authors want to say....Also the rest of the MS can still use serious English editing.

Extra: in the introduction the authors now mention (line 70-71) first that “in the present study the binding mode of doxepin..” which refers to their earlier seminal H1R work. Should this read as “in the earlier work/study”?? The authors then report on an active H1R structure, but fail to report on the recently reported inactive H2R and H3R structures.

Extra: line 155/252....is HA supposed to have an H-bond with its amine group with Asp94? Why not ionic as we assume for most aminergic GPCRs? Also, the imidazole ring is here supposed to interact with Glu residue via H-bonding. In line 238, the same interaction is mentioned as being an “ionic association”. The authors should align their statements.

Similar remarks are made for the interaction of OUP with receptor and might need adjustment. These data have now been added in the revised manuscript, but the methods of the docking are not described and the legend of extended data fig. 9 is not adjusted (see also point 10). This needs correction.

Reviewer #3 (Remarks to the Author):

The authors have addressed most of my comments and concerns adequately. I acknowledge that some of my suggestions may not be demonstrated within a reasonable time frame. I do not have any further comments.

Reviewer #4 (Remarks to the Author):

The authors have satisfied all concerns.

Reviewer #5 (Remarks to the Author):

In this paper authors provide cryo-electron microscopy structures of the histamine H4 receptor with agonists histamine and imetit. The structures were stabilized by binding to Gq protein which is not native to H4R but proved to be effective in stabilization of the receptor. The obtained structures are of

good quality and provide valuable information on agonist binding and possible sources of selectivity. Since this is the first structure of H4 receptor the value of this contribution is significant. The paper is well illustrated, both in the main part and in supplementary, and the obtained results, including large set of mutagenesis data, are very well and extensively discussed. There are also many comparisons with existing structures of histamine receptors which provide interesting conclusions on selectivity and also on the activation mechanism of H4R.

Although a discussion on mutagenesis data is rather extensive there is still something to be explained. The mutants Q347A and Q347G are the only mutants which activity increases compared to WT in case of activating by imetit. As mentioned in the text the imidazole group of imetit does not interact with Q347, so some explanation of such effect would be beneficial. Maybe some molecular modeling could be done to observe rearrangements of residues in the binding pocket of H4R for the above mutants.

The second case is the Y319A mutant. Decrease of H4 receptor activity is much smaller in case of imetit than for histamine although imetit forms pi-cation interactions with Y319, but no such binding exists for histamine.

REVIEWER COMMENTS

Below we provide our replies to the reviewers' comments in blue, following the reviewers' remarks in black.

Reviewer #1 (Remarks to the Author):

The authors addressed my questions, and the manuscript has been greatly improved. I can now support its publication in Nature Communications.

Thank you for your comment. We appreciate your inputs that helped us in improving the manuscript.

Reviewer #2 (Remarks to the Author):

In the present revision Im et al. present an updated version of their earlier work on H4R-agonist cryo-EM studies. It is good to see that a fair number of the comments have been addressed and that e.g. the position of the toggle switch has now been corrected, details on constructs and “undetermined parts of the structures” are now available.

Yet, also in the present version still a number of details on methodology/statistics etc are lacking. Moreover, the MS has been seriously rewritten, which at some parts raises again some issues. Also, a number of points have not been properly addressed and need imo further attention.

Thank you for your valuable input. In this new version, we have addressed the statistical analysis and methodology aspects per your suggestions.

Major comments

1. Introduction has been improved with references to the earlier work of Black/Arrang etc. Yet, the statement on the therapeutic use of pitolisant needs at least one reference. Also, it seems to me that the statements in line 80-82 have not been removed, as mentioned in the rebuttal?

Thank you for your comment. We have added a citation for the therapeutic use of pitolisant⁷ (Lines 54–57). In addition, we have removed the expressions you pointed out and summarized them with new simpler expressions (Lines 80–81).

2. a) it is nice to see the details on the miniG protein construct used.

Thank you for your comment.

b) the authors also claim that the structure with an unnatural G-protein is useful (despite the fact that G_q can hardly be activated by H4R in the hands of the authors themselves as indicated at 2d), by stating that “the protein is definitely in an active conformation”, without providing the details why they state this. I think the reader needs a bit of help to understand the stat value of the present complexes and that such a statement should be carefully discussed in the discussion of the MS.

Thank you for taking the advice. As you mentioned, G_q signal was not measurable in our study. However, previous research has demonstrated weak G_q coupling²². Moreover, we believe that the tethering system employed in this study would enhance the coupling with G_q by effectively bringing the GPCR and the trimeric G protein in close proximity. We have included this information in the Discussion section (Lines 356–361).

c) it is nice to see the extra data in extended fig 1, but no methods can be found for the experiments in fig. 1B for the H4R-G_i complex. Also, from the gel one can deduce that there is also another difference between the 2 efforts...the use of Nb35. In the paper it would be good to explicitly mention where the Nb35 is binding to the complex (and also for the svFv16). Did the authors try the same approach as with “the artificial G_q, that was based on the miniGs scaffold etc”? Did the authors try an artificial G_i, based on the same approach??

Thank you for your suggestion. We have added a method for H4R-G_i complex in Methods section (Lines 591–593, 628–630). As shown in the newly added method, Nb35 is not used for expression. The G_i used here is not the miniG_s that formed the backbone of G_q, but the G_i itself (with the dominant negative mutations), which Nb35 does not bind to, and therefore is not used in this experiment. So you are correct that you do not see a band corresponding to Nb35 in the gel. If we

tried using artificial G_i (based on the miniG_s scaffold) as you say, we might get the structure of the complex, but we did not attempt it in this experiment. Moreover, the binding sites of Nb35 and scFv16 have already been described in the text (Lines 99–102).

d) the authors should state clearly in the MS that they have been unable to measure Gq signaling with the WT H4R and discuss this in the context of their H4R complexes.

Thank you for your suggestion. We have added the relevant content to the manuscript as answered in b) (Lines 356–361).

3. good to see the details mentioned now

Thank you for your comment.

4. Still open comment: earlier paper(s) that have already been analyzing some of the H4R mutants in this paper are not mentioned. I do not appreciate this attitude. Proper referencing is good academic ethics.

Thank you for your comment. As you mentioned, it is appropriate to refer to and cite previous research in mutant analysis. We have added it to the revised manuscript (Lines 147–149).

With respect to the analysis of the mutants, the authors made some efforts, but I still have issues with the use of these data to support the cryo-EM data....The methods used to detect the expression are lacking (like for a fair number of other additions). What was used to detect functional expression? I presume some kind of antibody?? But which one and how, especially as there has been a discussion in the field on the use of commercial H4R antibodies? Also, the authors state “no mutants with significantly decreased cell surface expression were found”. Looking at the new figures Ext data 4a, I do not see any statistical analysis. The authors should add a statistical analysis to the figure and to the methods....and to me it seems that a fair number of mutants have a significantly decreased expression (even clear without statistical analysis) and I do not understand why the authors use the statement, mentioned above.

No statistical analyses were performed on the cell surface expression of each mutant. Therefore, "no mutants with significantly decreased cell surface expression were found" is incorrect. We have added the method used to detect the expression level each H₄R mutant (Lines 714–724). Regarding the expression levels of mutants and the results of TGF α shedding assay, T99A and N147A were found to have decreased expression levels compared to those of WT; EC₅₀ and E_{max} were measured. By contrast, the expression level of Y95A was increased compared to that of WT, but EC₅₀ and E_{max} were not measured. The results indicate that the result of TGF α shedding assay does not depend greatly on differences in expression levels, and there is no problem in using these data as support for the cryo-EM structures.

	Histamine				Imetit			
	pEC50 mean \pm SEM	E _{max} mean \pm SEM	EC50 (nM)	fold of change	pEC50 mean \pm SEM	E _{max} mean \pm SEM	EC50 (nM)	fold of change
WT	7.28 \pm 0.15	19.85 \pm 1.17	52	1	7.54 \pm 0.10	15.06 \pm 0.55	28.7	1
D94A	ND	ND	ND	+++	ND	ND	ND	+++
Y95A	ND	ND	ND	+++	ND	ND	ND	+++
C98A	ND	ND	ND	+++	6.31 \pm 0.09	17.02 \pm 0.91	487.4	17.0
T99A	6.75 \pm 0.29	25.81 \pm 3.94	176.2	3.4	7.12 \pm 0.11	13.87 \pm 0.79	76.2	2.7
N147A	6.35 \pm 0.14	14.15 \pm 1.26	444.7	8.6	6.83 \pm 0.14	19.88 \pm 1.35	147.1	5.1
N147W	ND	ND	ND	+++	6.24 \pm 0.07	17.97 \pm 0.80	579.9	20.2
N147Y	7.09 \pm 0.06	12.63 \pm 0.41	81.3	1.6	6.51 \pm 0.35	1.8 \pm 2.93 \pm 2.22	307.5	10.7
E182A	ND	ND	ND	+++	ND	ND	ND	+++
E182N	ND	ND	ND	+++	ND	ND	ND	+++
Y319A	ND	ND	ND	+++	6.78 \pm 0.18	16.98 \pm 1.39	164.9	5.7
F344A	ND	ND	ND	+++	ND	ND	ND	+++
F344I	ND	ND	ND	+++	ND	ND	ND	+++
Q347A	6.09 \pm 0.13	10.03 \pm 1.04	817.2	15.7	7.92 \pm 0.16	20.45 \pm 0.92	12	0.4
W348A	ND	ND	ND	+++	ND	ND	ND	+++
W348Y	ND	ND	ND	+++	ND	ND	ND	+++

ND, EC50 or E_{max} values cannot be detected due to their low signal; +++, Exceedingly high fold of change due to the undetectable or very low agonist activity

5. although fig. 2f does not show a huge effect of the Tyr319Ala mutation, the authors report that the EC₅₀ is changed approximately 5-fold. First of all, there are other changes that are \pm 5-fold (e.g. Q349G). Does that also highlight an important interaction? More importantly, there is no statistical analysis of the data in the Supp data Table 1. The authors should add such an analysis for all mutants and then discuss the changes observed. Also, would a lower expression of the mutant explain the 5-fold shift perhaps?

Thank you for considering the suggestion. In response to it, we performed statistical analyses on all data in Supplementary Data Table 1 and incorporated the results. We have also described how this was done (Lines 726–730). An important aspect of this study is that the pharmacological outcomes can be rationally discussed based on the structure. Naturally, it was challenging to explain all pharmacological results solely from our structures. Therefore, in this paper, we

considered that mutants inducing approximately a 5-fold change would have some effect on ligand binding. For example, we believe Q349G/A enhances the signal of imetit. The extended side chain conformation of Gln347 is thought to restrict the conformation of Trp316. Mutation of Gln347 may remove this restriction, providing flexibility to the rotamer of Trp316 and enabling it to approach Imetit more closely, thereby strengthening the interaction. In the case of histamine, the interaction between Gln347 and the imidazole ring was observed; therefore, we believe the mutation reduces the activity. We have added this point to the revised manuscript (Lines 293–299).

6. that authors corrected the position of the important 6.48 residue.

Thank you for your comment.

7. see point 2 on the mutations

Thank you for your comment.

8. see 6

Thank you for your comment.

9. the authors followed the earlier suggestion.

Thank you for your comment.

10. JNJ7777120 docking in the active structure has now been added, which I suggested in the first round of review instead of the somewhat artificial focus on constitutive activity (and that was not shown experimentally). As JNJ is potentially a biased agonist, this approach is perhaps not as uncommon as it may seem, but the present work is not positioned in the present paper like this.

Thank you for pointing this out. As you mentioned, JNJ-777120 is definitely a very important ligand in the study of H₄R. We also think it is an interesting compound with activity as a biased

agonist to β -arrestin. We have added this pharmacological feature in the revised manuscript (Lines 388–389).

Now it is really odd to read that an antagonist is being docked into an active structure. I also suggest that the authors do MD studies to allow the system to equilibrate to a potentially inactive structure? Also, how do the present results compare to docking in e.g. an inactive H4R model present at GPCRdb and the Alphafold model? Moreover, there are some mutagenesis data in the literature on JNJ binding and the current pose should be considered in the context of such data.

In this study, we have attempted to predict where this compound might bind using docking simulations. As you mentioned, the current docking model is a simulation based on the active form structure we have determined. We should have done this with the inactive structure, but since that structure is unknown at this time, we are utilizing the structural model we have obtained to predict the pocket (residue) that JNJ-777120 may use. In existing reports, mutant experiments have demonstrated that Asn147, Phe169, Leu175, and Ser179 affect the activity of JNJ-777120. Our predicted docking analysis also shows that the chloroindole ring of JNJ-777120 is adjacent to these residues, and since this region is an H4R-specific sub-pocket not found in H3R, our assertion using this model is consistent. We cite these mutation experimental data in the revised manuscript (Lines 398–400). A comparison of the inactive H4R structure in the proposed GPCRdb and AF2 model with the present docking structure is shown below. The arrangement of the residues constituting the sub-pocket, circled by the black dotted line, is almost identical in the two models, suggesting that the chloroindole ring may bind similarly in the inactive structure.

A minor point: methods of the computational approach cannot be found and the legend of the figure is not mentioning the JNJ docking pose.

Thank you for pointing this out. The docking method is described in Methods - Molecular docking (Lines 678–683). Please check it. We apologize that the figure legend was not updated and we have corrected it (Lines 897–898).

Extra: Abstract need some editing (histamine agonist do not exist...also, one can not talk about “subtype selectivity of histamine receptors”, although I understand what the authors want to say....Also the rest of the MS can still use serious English editing.

Thank you for your comment. Regarding the selectivity for histamine receptors, we discussed the difference between H₁R and H₄R in histamine binding, the presence or absence of an aromatic slot (H₁R, H₂R/H₃R, H₄R) that affects imetit binding, and the possibility that H₄R-specific ligands such as JNJ-777120 can differentiate from H₃R by utilizing the H₄R-specific sub-pocket. Therefore, we would like to adopt this phrase. However, our primary focus in this study is on the structure of agonist binding. Therefore, we have removed the mention of the development of CID drug that is an antagonist of H₄R in the Abstract (Lines 45–46).

Extra: in the introduction the authors now mention (line 70-71) first that “in the present study the binding mode of doxepin..” which refers to their earlier seminal H1R work. Should this read as “in the earlier work/study”?? The authors then report on an active H1R structure, but fail to report on the recently reported inactive H2R and H3R structures.

Thank you for your feedback. There was a mistake in the use of the word; we will adopt your suggestion of “earlier” instead of “present” (Line 70). Furthermore, the structure of the inactive form of H₃R was published after we submitted this paper while it was under review, and we addressed that in the first revision (Lines 337–338). We have added the new inactive structure of H₂R in the revised manuscript.

Extra: line 155/252....is HA supposed to have an H-bond with its amine group with Asp94? Why not ionic as we assume for most aminergic GPCRs? Also, the imidazole ring is here supposed to interact with Glu residue via H-bonding. In line 238, the same interaction is mentioned as being an “ionic association”. The authors should align their statements.

Thank you for pointing this out. There was an incorrect representation of the interaction. As you mentioned, Asp94 forms an ionic interaction with the ethylamine moiety of histamine. Glu182 is also considered to form an ionic pair with N⁺ of imidazole. We have changed the incorrect wording in the relevant sections of the manuscript (Lines 155–157, Line 254).

Similar remarks are made for the interaction of OUP with receptor and might need adjustment. These data have now been added in the revised manuscript, but the methods of the docking are not described and the legend of extended data fig. 9 is not adjusted (see also point 10). This needs correction.

Thank you for your suggestion. As we answered in 10), the docking method is described in Lines 678–683. We have adjusted the legend for Extended Data Figure 9 (Lines 897–898).

Reviewer #3 (Remarks to the Author):

The authors have addressed most of my comments and concerns adequately. I acknowledge that some of my suggestions may not be demonstrated within a reasonable time frame. I do not have any further comments.

Thank you for your valuable feedback. We appreciate your support in improving our manuscript.

Reviewer #4 (Remarks to the Author):

The authors have satisfied all concerns.

Thank you for your comment. We appreciate your assistance in enhancing the manuscript.

Reviewer #5 (Remarks to the Author):

In this paper authors provide cryo-electron microscopy structures of the histamine H4 receptor with agonists histamine and imetit. The structures were stabilized by binding to Gq protein which is not native to H4R but proved to be effective in stabilization of the receptor. The obtained structures are of good quality and provide valuable information on agonist binding and possible sources of selectivity. Since this is the first structure of H4 receptor the value of this contribution is significant. The paper is well illustrated, both in the main part and in supplementary, and the obtained results, including large set of mutagenesis data, are very well and extensively discussed. There are also many comparisons with existing structures of histamine receptors which provide interesting conclusions on selectivity and also on the activation mechanism of H4R.

We wish to express our appreciation to the reviewer for these insightful comments, which have helped us to significantly improve the paper. We have addressed each of your comments below.

Although a discussion on mutagenesis data is rather extensive there is still something to be explained. The mutants Q347A and Q347G are the only mutants which activity increases compared to WT in case of activating by imetit. As mentioned in the text the imidazole group of imetit does not interact with Q347, so some explanation of such effect would be beneficial. Maybe

some molecular modeling could be done to observe rearrangements of residues in the binding pocket of H4R for the above mutants.

Thank you for your comment. As you mentioned, Q347A/G are the only mutants that show improved activity with imetit. Unlike histamine, the imidazole ring of imetit does not interact with Q347. Therefore, while mutations at this site lead to reduced activity with histamine, they do not have the same effect with imetit. As you pointed out, introducing mutations at this residue is expected to cause structural changes in the surrounding pocket or specific residues. We particularly focus on Trp316. As mentioned in the “Activation Mechanism of H4R” section (Lines 337-353), it is believed that the extended side chain conformation of Gln347 restricts the conformation of Trp316 (Toggle switch). Mutation of Gln347 may remove this restriction, providing flexibility to the rotamer of Trp316 and thereby potentially strengthening the interaction with imetit. We have added this information to the revised manuscript (Lines 293-299).

The second case is the Y319A mutant. Decrease of H4 receptor activity is much smaller in case of imetit than for histamine although imetit forms pi-cation interactions with Y319, but no such binding exists for histamine.

As you pointed out, the difference in activity between WT and Y319A is much smaller for imetit

than for histamine. This is likely due to the charge state of the isothioureia moiety of imetit (Lines 206–209). The entire site is protonated and charged, suggesting that it forms multiple interactions with the entire pocket. In particular, it forms hydrogen bonds (ionic interactions) with Asp94 and Phe344, making it difficult to induce a significant decrease in activity with a single Y319A mutation. On the other hand, we believe that the imidazole ring of histamine forms a weak pi–pi interaction with Y319, which is likely responsible for the decreased activity in the mutant.